# A link between STK signalling and capsular polysaccharide synthesis in *Streptococcus suis*

Jinsheng Tang [1], Mengru Guo [1], Min Chen[1], Bin Xu[2], Tingting Ran[3], Weiwu Wang[3], Zhe Ma[1,4], Huixing Lin[1,4] & Hongjie Fan [1,4] ✉

Synthesis of capsular polysaccharide (CPS), an important virulence factor of pathogenic bacteria, is modulated by the CpsBCD phosphoregulatory system in Streptococcus. Serine/threonine kinases (STKs, e.g. Stk1) can also regulate CPS synthesis, but the underlying mechanisms are unclear. Here, we identify a protein (CcpS) that is phosphorylated by Stk1 and modulates the activity of phosphatase CpsB in *Streptococcus suis*, thus linking Stk1 to CPS synthesis. The crystal structure of CcpS shows an intrinsically disordered region at its N-terminus, including two threonine residues that are phosphorylated by Stk1. The activity of phosphatase CpsB is inhibited when bound to non-phosphorylated CcpS. Thus, CcpS modulates the activity of phosphatase CpsB thereby altering CpsD phosphorylation, which in turn modulates the expression of the Wzx-Wzy pathway and thus CPS production.

Many bacteria can retain the capsular polysaccharide on the cell surface to form a capsule or release it to the surrounding environment[1,2]. CPS is an important virulence factor of bacterial pathogens, which can help bacteria deal with dramatic environmental changes and against the host immune system, including complement deposition and opsonophagocytosis[2,3]. Capsules also can promote intracellular survival and translocation[4], transmission, and persistence of bacteria[5]. It has recently been reported that capsules can help bacteria block the killing of antimicrobial peptides[6,7]. However, how bacteria effectively regulate capsule production to promote them survive in different environmental or host niches is poorly understood.

Although the structures and components of bacterial CPS are quite different, there are two main pathways, the ABC-transporter dependent pathway and Wzx-Wzy pathway for CPS synthesis, the latter being the most widespread, and it is found in both gram-negative and gram-positive bacteria[2]. The Wzx-Wzy system is used to build many capsules, which protect pathogens against host innate immune defenses, in which CPS synthesis-related genes are organized into a large operon[8]. Cocci, like *S. suis*, encodes the first four genes in the *cps*

locus, *cpsABCD*, which are broadly conserved and are important in the modulation of capsule synthesis[8,9]. The *cpsA* gene product is related to attaching the CPS to the cell wall peptidoglycan of bacteria[10]. It has been reported that CpsBCD forms a phosphoregulatory system to regulate CPS production[11–13]. Further, immediately downstream genes encode the initiating phosphoglycosyl transferase and other late-stage enzymes[8,14]. The level of CPS is regulated in several ways, including the change in the transcription level of the *cps* gene[15], the significance of enzymes related to CPS synthesis[16], and the control of posttranslational modification by the CpsBCD phosphoregulatory system[11,12]. The latter is the most complex, and its regulation mechanism has not been fully revealed in Gram-positive and Gram-negative bacteria.

*Escherichia coli* tyrosine kinase Wzc shares no resemblance with eukaryotic counterparts and belongs to the bacterial tyrosine kinase family (BY)[17], which contains Walker A and B motifs, as well as a tyrosine-rich tail presenting several residues for phosphorylation[18]. The difference is that most Gram-positive bacteria BY kinases are composed of two independent proteins: CpsC and CpsD[13,17,19]. CpsC is a membrane protein containing two short cytoplasmic regions at the

[1]MOE Joint International Research Laboratory of Animal Health and Food Safety, College of Veterinary Medicine, Nanjing Agricultural University, Nanjing 210095, China. [2]National Research Center of Veterinary Biologicals Engineering and Technology, Jiangsu Academy of Agricultural Sciences, Nanjing 210000, China. [3]Department of Microbiology, College of Life Sciences, Nanjing Agricultural University, Nanjing 210095, China. [4]Jiangsu Coinnovation Center for Prevention and Control of Important Animal Infectious Diseases and Zoonoses, Yangzhou 225009, China. ✉e-mail: fhj@njau.edu.cn

amino and carboxy terminals, and the carboxy-terminal sequence is essential for the activity of CpsD auto-kinase[20], while CpsD-P can be dephosphorylated by CpsB, which is a tyrosine-protein phosphatase[21]. Studies have reported that abnormal phosphorylation of CpsD is closely associated with the production of bacterial CPS[12,22,23], but how the CpsBCD phosphoregulatory system regulates the CpsD phosphorylation level, thereby modulating CPS productions and promotes the survival of bacteria against stress, especially when bacteria cope with harsh environment or host immune defense, is largely unknown.

How bacteria sense extracellular stimuli and respond to the environmental changes around them is a fundamental question of bacterial physiology. Eukaryotic-like serine/threonine kinase (eSTK) is the new hallmark of bacterial phosphosignaling, controlling almost all aspects of bacterial physiology[24,25]. There is evidence that eSTK can sense PG fragments and induce germination[26], suggesting the ability of bacteria to sense and respond to adverse environmental conditions. It has been reported that many processes of cell wall synthesis are regulated by eSTK[27–29], which also plays an important role in cell division, morphology, and virulence of bacteria[29–32]. Various substrates of eSTK have been identified in recent years, and many eSTK substrates were thought to play important roles in various biological processes of bacteria[24,25]. For example, GpsB, as an adapter protein, coordinates PG synthase and other cell wall synthase to promote cell division and elongation[33]. As a substrate of eSTK, GpsB can be phosphorylated and negatively regulate eSTK kinase activity[34,35]. Interestingly, previous studies showed that eSTK could affect the processes of CPS and PG synthesis simultaneously[36,37]. However, how eSTK coordinates the two biological processes, especially how to regulate the production of CPS when the environment changes sharply around bacteria, is poorly understood.

Here, we provide evidence that bacteria have evolved sophisticated mechanisms for building a linkage between signal transduction by Stk1/Stp1 system and the Wzx-Wzy pathway. We identify a small protein, CcpS, which can be specifically phosphorylated by Stk1/Stp1 system and demonstrate that the Stk1-CcpS axis is active. CcpS plays an important role in CPS synthesis in *S. suis*, which is bound to the Wzx-Wzy pathway protein CpsB. The structure of *S. suis* CcpS solved by X-ray crystallography showed a disordered region at the N-terminus of CcpS. Interestingly, two phosphorylation residues Thr4 and Thr7 are included in intrinsically disordered regions (IDRs) of CcpS. Importantly, we report that CcpS is functionally dependent on IDRs, which can be tuned by phosphorylation, and homo-dimer structure is essential for its function. Non-phosphorylated CcpS has a higher affinity to CpsB and inhibits its activity, thereby modulating CpsD phosphorylation, and, thus, CPS production. Our results are consistent with the idea that the Stk1-CcpS axis probably constitutes a widespread signalling and regulation mechanism in bacteria due to the unique structure of CcpS, and its phosphorylation are conserved.

## Results

### Thr-phosphorylation of CcpS protein is specifically mediated by Stk1/Stp1

*S. suis* is an important zoonotic pathogen that can cause Streptococcal toxic shock-like syndrome (STSLS), and meningoencephalitis poses a serious threat to human health[38]. We found that *S. suis* also encodes a serine/threonine kinase Stk1 and its cognate serine/threonine phosphatase Stp1. In silico analysis indicated that Stk1 includes the typical N-terminal STK kinase domain in the cytoplasm (1–346 amino acids) and the C-terminal PASTA domain (367–664 amino acids) in the periplasm separated by a transmembrane region (347–366 amino acids) (Supplementary Fig. 1a, b), which was similar to Stk1 homologs in other bacteria[25]. In a previous study, we identified a potential Stk1-specific substrate hypothetical protein RS00400 (renamed CcpS)[39] by phosphoproteomic analysis. Strikingly, we detected a phosphorylated version of CcpS-specific peptides, and MS/MS analysis showed that

CcpS phosphorylation occured on two threonine residues Thr4 and Thr7 (Supplementary Fig. 1c). The gene *ccpS* and the other three genes *rs00395*, *rs00405*, and *rs00410* encode three hypothetical proteins, respectively, with unknown functions in a putative operon (Fig. 1a). Intriguingly, gene prediction indicated that *ccpS*, encoding an IreB-like family regulatory phosphoprotein that can influence antimicrobial resistance in *Enterococcus faecalis*[40], the function of which is unknown.

Bacterial two-hybrid assays in *E. coli* showed that CcpS and Stk1 form an interaction complex and self-interact (Fig. 1b), which suggested CcpS and Stk1 can keep physiological activity in a heterologous host. Therefore, to further confirm whether Stk1 can phosphorylate CcpS, we first constructed the co-expression system of CcpS and Stk1 in *E. coli* and indicated that the phosphorylation of CcpS was strictly dependent on the presence of Stk1 (Fig. 1c, d). Subsequently, to validate whether Stk1 can directly phosphorylate CcpS, the purified recombinant protein rCcpS was incubated in the presence or absence of truncated protein rStk1$_{1-346}$ in vitro. The results showed that CcpS can be phosphorylated in the presence of rStk1$_{1-346}$ but not in the absence of rStk1$_{1-346}$ (Supplementary Fig. 1d). Further immunoprecipitation (IP) analysis showed that CcpS is only phosphorylated by Stk1 in *S. suis*, because in the absence of CcpS or Stk1, no CcpS phosphorylation can be detected in *S. suis* (Fig. 1e).

Then, to determine whether the phosphorylation of CcpS occurs specifically at residues Thr4 and Thr7, we carried out a series of experiments. We first tested the influence of CcpS variants on interaction with Stk1, and found that CcpS variants CcpS-T4ET7E (mimics phosphorylated Thr residues) and CcpS-T4VT7V (mimics non-phosphorylated Thr residues) still formed interaction complexes with Stk1 (Supplementary Fig. 1e). Subsequently, we incubated rCcpS and its variants with rStk1$_{1-346}$ in vitro and resolved the proteins into Phos-tag gel[41]. The results showed that both residues Thr4 and Thr7 can be phosphorylated by Stk1 in vitro (Supplementary Fig. 1f). Furthermore, heterologous co-expression of CcpS and its variants in the presence of Stk1 in *E. coli* also indicated that the Thr4 and Thr7 residues of CcpS can be phosphorylated by Stk1, respectively (Supplementary Fig. 1g). Importantly, IP analysis of CcpS phosphorylation in *S. suis* WT (wild type) strain and mutants strain showed that CcpS was phosphorylated only on two specific threonine residues, Thr4 and Thr7 (Fig. 1f). Next, we tested whether the Stk1 cognate phosphatase Stp1[42] can remove phosphoryl groups from CcpS-P specifically in vitro. The enzymatic activity analysis indicated that Stp1 is a phosphatase with divalent manganese ion-dependent activity (Supplementary Fig. 1h). Further, phosphatase assays analysis showed that Stp1 rather than CpsB[11], which has a tyrosine-protein phosphatase activity, can specifically dephosphorylate the CcpS-P (Fig. 1g, h). Altogether, these experiments indicated that Stk1 and CcpS form an interaction complex and the kinase Stk1 specifically phosphorylates CcpS at residues Thr4 and Thr7. In contrast, Stp1 can reversibly dephosphorylate CcpS-P.

### CcpS phosphorylation in *S. suis* was regulated by multiple extracellular stimuli

To explore the physiological significance of CcpS phosphorylation in the life course of *S. suis* pathogen, we subjected *S. suis* WT strain to several biological and chemical stressors, and thus assessed the phosphorylation levels of CcpS. We first tested the CcpS phosphorylation level of cells in the normal growth of *S. suis*. Notably, there was a significant increase in CcpS phosphorylation levels of cells in the logarithmic phase (Fig. 2a, b), which suggested that CcpS phosphorylation may participate in important cellular processes of bacteria. It was reported that CcpS homolog IreB phosphorylation in *E. faecalis* was related to nutrient limitation[43]. Therefore, we also treated the *S. suis* cells with continuous starvation in PBS. The results showed that the CcpS phosphorylation level of cells decreased rapidly and maintained at a low level over time (Supplementary Fig. 2a, b), which was similar to

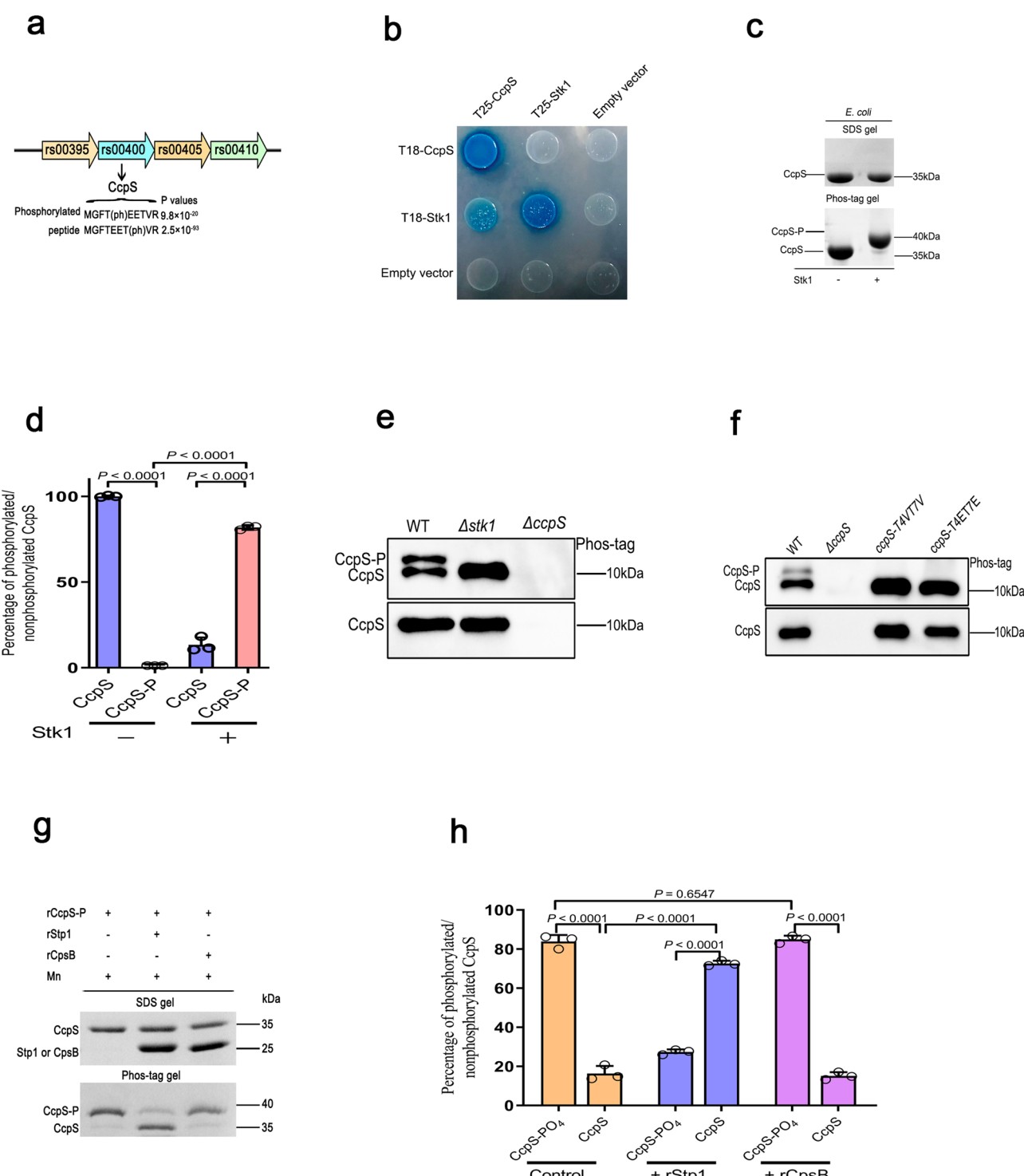

**Fig. 1 | The CcpS phosphorylation is modulated by the Stk1/Stp1 system. a** The schematic shows the genetic context of *rs00400* (*ccpS*) and the phosphorylation possibility of detected phosphorylated CcpS peptides, which was calculated by Maxquant (version No. 1.5.2.17), *n* = 3 independent biological replicates. **b** The interaction between Stk1 and CcpS was verified by a bacterial two-hybrid assay. Blue colony formation suggests that a direct interaction occurs. **c** CcpS-GST was expressed in the absence or presence of ectopic expression of Stk1 in heterologous host *E. coli* cells, GST agarose beads were used to pull down CcpS-GST, and the samples were loaded onto Phos-tag gel to analyze the phosphorylation of CcpS, samples also were loaded onto a standard SDS gel as a loading control and Coomassie stain. **d** Bar graph shows the percentage of phosphorylated and non-phosphorylated CcpS in *E. coli* cells in the absence or presence of ectopic expression of Stk1 (density analysis for **c**). Data represent mean ± SD from *n* = 3 biologically independent experiments. **e, f** IP analysis of CcpS phosphorylation in *S. suis* in

the presence or absence of Stk1 (**e**), and in *S. suis* strain *ccpS-T4ET7E* and *ccpS-T4VT7V*, which encodes the phospho-mimetic variant CcpS-T4ET7E and phospho-ablative variant CcpS-T4VT7V, respectively (**f**). Western blot of the immunoprecipitate using anti-CcpS antibodies was analyzed on Phos-tag gel for CcpS phosphorylation, and on a standard SDS gel acting as a loading control. **g** The dephosphorylation of CcpS by Stp1 was analyzed in vitro. The recombinant protein rStp1 (r means recombinant protein) dephosphorylate rCcpS-P was detected by Phos-tag gel and a standard SDS gel, and the recombinant protein rCpsB as a control and Coomassie stain. **h** Bar graph shows the percentage of phosphorylated and non-phosphorylated CcpS in the absence or presence of Stp1 or CpsB (density analysis for **g**). Data represent mean ± SD from *n* = 3 biologically independent experiments. Statistical difference: unpaired two-tailed student's *t*-test (**d, h**). *P* values <0.05 indicate significant differences. Source data are provided as a Source Data file.

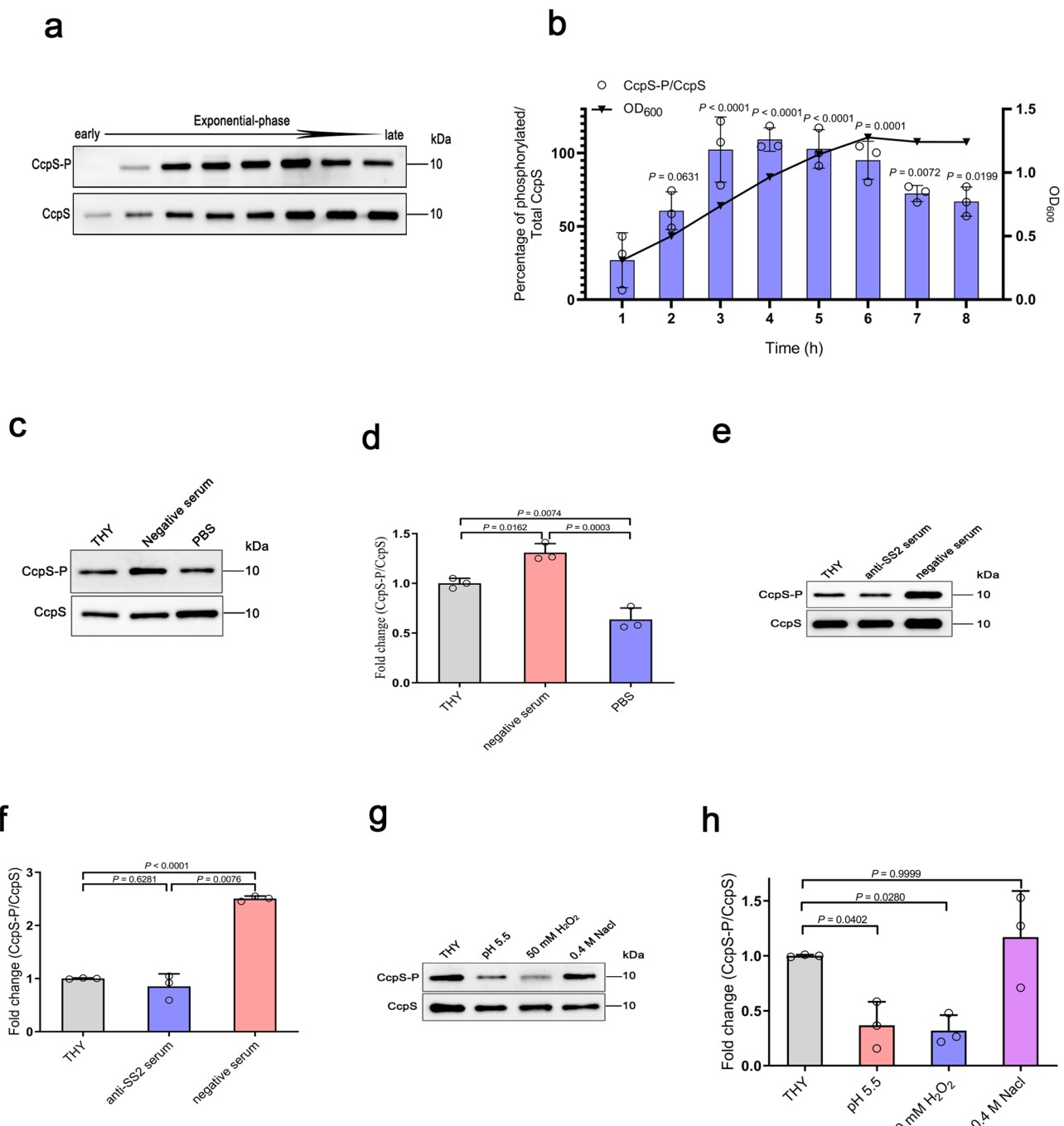

**Fig. 2 | CcpS phosphorylation participates in various stress response.** IP analysis of *S. suis* CcpS phosphorylation using anti-CcpS beads from whole-cell lysates of cells with normal growth or various treatments. Western blot of immunoprecipitate analysed on an SDS gel using anti-CcpS antibodies for CcpS and anti-phosphothreonine antibody for CcpS-P. **a** Cells from different growth stages, cells were harvested at the indicated time from early to late exponential-phase. **c** Exponentially growing cells were treated with starvation in PBS or nutrient medium (THY supplemented with negative serum for *S. suis* at a ratio of 1:1) for 20 min. **e** Exponentially growing cells were treated with THY supplemented with anti-serotype 2 antisera at a ratio of 1:1 or nutrient medium for 20 min. **g** Exponentially growing cells were treated with pH 5.5 THY adjusted by hydrochloric acid, THY supplemented with 0.4 M NaCl, and THY supplemented with 50 mM $H_2O_2$ for 20 min. **b, d, f, h** Bar graphs showed the percentage of phosphorylated and total CcpS in different treatment groups (Density analysis for **a, c, e, g**, respectively). For **b, d, f, h**, Data represent mean ± SD from $n = 3$ biologically independent experiments. Statistical difference: one-way ANOVA followed by Bonferroni or Tukey's post-tests. *P* values <0.05 indicate significant differences. Source data are provided as a Source Data file.

the previously reported that CcpS homolog IreB phosphorylation decreased in PBS treatment[43], while the total CcpS level no change. In addition, we found that starvation in PBS has no effect on cell survival in a short time (Supplementary Fig. 2c). Here, we speculated that the accessibility of cells to nutrients may disturb the Stk1-CcpS axis in *S. suis*. To test this hypothesis, we used negative serum (means negative serum

for *S. suis*) as a nutrient medium and sterile PBS as starvation. As expected, the CcpS phosphorylation levels of cells rapidly increased in a nutrient medium, while the CcpS phosphorylation levels decreased significantly in starvation compared to cells in the THY medium (Fig. 2c, d). Notably, Stk1 kinase activity analysis showed that the Stk1 phosphorylation levels of cells increased rapidly in a nutrient medium

but decreased significantly in starvation, which was in agreement with CcpS phosphorylation (Fig. 2c, d and Supplementary Fig. 2d, e). And this result further supported that the CcpS phosphorylation level of cells depends on Stk1 kinase activity.

To further understand the dynamic changes of CcpS phosphorylation in *S. suis* against extracellular stimuli, we subjected WT strain cells to several cellular stresses in vitro, to some extent, which can simulate host factor in niches. Interestingly, we found that the CcpS phosphorylation level of cells did not change when the cells were treated with anti-serotype 2 antiserum compared to cells in THY medium (Fig. 2e, f), while CcpS phosphorylation level of cells increased significantly in a nutrient medium, which suggested that the activation or inhibition of the CcpS phosphorylation mediated by Stk1/Stp1 system depends on cells against different host factors. In addition, we also subjected WT strain cells into several biochemical stressors in vitro, the results showed that the CcpS phosphorylation level of cells decreased significantly in acidic environments and hydrogen peroxide, while osmotic pressure has no effect on CcpS phosphorylation (Fig. 2g, h). Together, these results indicated that the Stk1-CcpS axis is active in the life course of *S. suis*. Therefore, we speculated that CcpS phosphorylation may be involved in some important cellular biological processes of *S. suis*.

## CcpS is closely associated with CPS production and phosphorylation is required for its activity

To explore the biological functions of CcpS and its phosphorylation events in cell physiology, we carried out a series of experiments to test the importance of the Stk1-CcpS system against several cellular stresses. We first tested the cell growth, and found that the *ΔccpS* mutant strain grew as fast as the WT strain when cultivated at 37 °C (Supplementary Fig. 3a). It has been reported that eSTK and its substrates were associated with cell wall metabolism in bacteria[27–29]. Thus, to test whether the Stk1-CcpS system affected peptidoglycan biosynthesis, we treated the *S. suis* WT strain and its mutants strain against the cephalosporin antibiotics. The results showed that CcpS and its phosphorylation have no effect on cephalosporin resistance (Supplementary Fig. 3b), while Stk1 depletion lowered cephalosporin resistance and *Δstk1ΔccpS* double mutant strain exhibited slightly enhanced resistance to cephalosporin compared to the *Δstk1* parent. These results were different from previous studies that the effect of Stk1 and CcpS homologs on cephalosporin resistance in *E. faecalis* and *Listeria monocytogenes*[40,44]. Furthermore, when cells lacked Stk1 expression, the rate of cell autolysis was faster than the WT strain, while CcpS and its phosphorylation have no effect on bacterial autolysis. However, there is no significant difference in bacterial autolysis between the WT strain and the *Δstk1ΔccpS* double mutant strain (Supplementary Fig. 3c). Interestingly, a similar phenomenon was also observed in *L. monocytogenes*[45]. In addition, previous studies showed that, phosphorylation of ReoM, which is a CcpS homolog, is essential for indirectly regulating the degradation of MurA in *L. monocytogenes*[44]. Therefore, to test whether CcpS phosphorylation has a similar function in *S. suis*, we analyzed homologous MurA1 and MurZ proteins levels in *S. suis ccpS-T4ET7E* mutant strain, *ccpS-T4VT7V* mutant strain, and WT strain. However, both MurA1 and MurZ proteins levels were present at WT strain levels in *ccpS-T4ET7E* and *ccpS-T4VT7V* mutant strains (Supplementary Fig. 3d–g), which suggested that the function of CcpS phosphorylation may not be mediated by regulated proteolysis. In addition, we observed that single *ΔmurA1* or *ΔmurZ* mutant strain has no significant effect on *S. suis* resistance to cephalosporin (Supplementary Fig. 3h). Therefore, these results also explained why the above phenotypes caused by CcpS deletion or its variants in *S. suis* were different from its homologs in *E. faecalis* and *Listeria monocytogenes*[40,44]. Furthermore, *ccpS* disruption did not cause an overall change in cell shape, and the overall cell shape of *ccpS-T4ET7E* and *ccpS-T4VT7V* mutant stains were similar with WT strain (Supplementary Fig. 3i, j). Thus, similar to other bacterial eSTK, Stk1 was also

associated with cell wall metabolism in *S. suis*. While CcpS seems to has some relationship with peptidoglycan biosynthesis and cephalosporin resistance in *S. suis*, but its phosphorylation function does not appear to be essential. CcpS may therefore have other functions in *S. suis*.

Interestingly, we found that *ΔccpS* mutant strain was easier to be swallowed by macrophage RAW264.7 than the WT strain, and ectopic expression of CcpS by a shuttle vector can recover the resist phagocytosis ability of bacteria (Supplementary Fig. 3k). In addition, the resist phagocytosis ability of the *Δstk1* mutant strain also increased significantly, while the possibility of *Δstk1ΔccpS* double mutant strain being engulfed by macrophages was higher than WT strain (Supplementary Fig. 3k). Thus, these results indicated that the function of Stk1-CcpS system is associated with *S. suis* resisting phagocytosis of macrophages. Notably, the *ccpS-T4VT7V* mutant strain also showed an increase in the resist phagocytosis ability of cells (Supplementary Fig. 3l), while the *ccpS-T4ET7E* mutant strain showed a similar resist phagocytosis ability compared with WT strain. This result indicated that the function of CcpS phosphorylation was also involved in *S. suis* resisting the phagocytosis of macrophages. It has been reported that CPS is an important virulence factor that helps bacteria to resist phagocytosis[1], and *S. suis* is also covered with a layer of thick capsular polysaccharide[46]. Furthermore, our results also showed that the resist phagocytosis ability of the non-encapsulated *Δcps* mutant strain significantly decreased compared to *S. suis* WT strain (Supplementary Fig. 3k), which suggested that CPS is also contributed to *S. suis* resisting phagocytosis of macrophages. Subsequently, to further determine whether the Stk1-CcpS system affects the CPS production in *S. suis*, the amount of CPS's sialic acid (SA)[46] presented on the surface of strains was quantified in bacterial extracts obtained by neuraminidase treatment[47]. We detected lower amounts of cell surface-associated CPS in *ΔccpS* mutant strain compared with the WT strain (Fig. 3a), while the amount of cell surface-associated CPS by the complemented strain was restored to WT strain levels. In addition, to examine the cell-free CPS (CPS released to the surrounding environment), serial dilutions of spent growth media were spotted on a PVDF membrane and probed with an anti-CPS antibody. The results showed that *ΔccpS* mutant strain decreased the amount of CPS in the medium compared with the WT strain and complemented strain restored the amount of cell-free CPS to WT strain levels (Fig. 3b, c). Interestingly, the *Δstk1* mutant strain showed a significant increase in the amount of cell-free CPS in the culture supernatant, while the *Δstk1ΔccpS* double mutants strain showed a decrease in the amount of cell-free CPS compared with the WT strain (Fig. 3d, e). These results indicated that the Stk1-CcpS system is required for CPS production in *S. suis*.

Interestingly, the *ccpS-T4VT7V* mutant strain showed a significant increase in the amount of cell-free CPS, while *ccpS-T4ET7E* mutant strain showed a similar cell-free CPS level compared with the WT strain (Fig. 3f, g). Importantly, the strain expressing the CcpS-T4VT7V variant also increased the amount of cell-free CPS, while the strain expressing CcpS-T4ET7E showed a similar cell-free CPS level compared with WT strain in a *Δstk1* mutant strain background (Supplementary Fig. 3m, n). These results suggested that the function of CcpS phosphorylation was also involved in *S. suis* CPS production. Furthermore, transmission electron microscope analysis of the cell capsule showed that the average thickness of the CPS layer decreased after the deletion of *ccpS*, but the strain *ccpS-T4VT7V* increased the average thickness of the CPS layer compared with the WT strain (Fig. 3h, i). Together, these results indicated that CcpS and Stk1 operated in the same biological pathways, and the Stk1-CcpS system was required for bacteria to resist phagocytosis and CPS production in *S. suis*. Importantly, CcpS activity was tuned by phosphorylation mediated by Stk1/Stp1 system. Thus, we renamed RS00400 as CcpS (Coordinator of capsular polysaccharide synthesis).

## CcpS interacts with CpsB and modulates its activity

In silico analysis of the CcpS sequence and functional domain predicted can not explain its regulation on CPS production in *S. suis*. Thus,

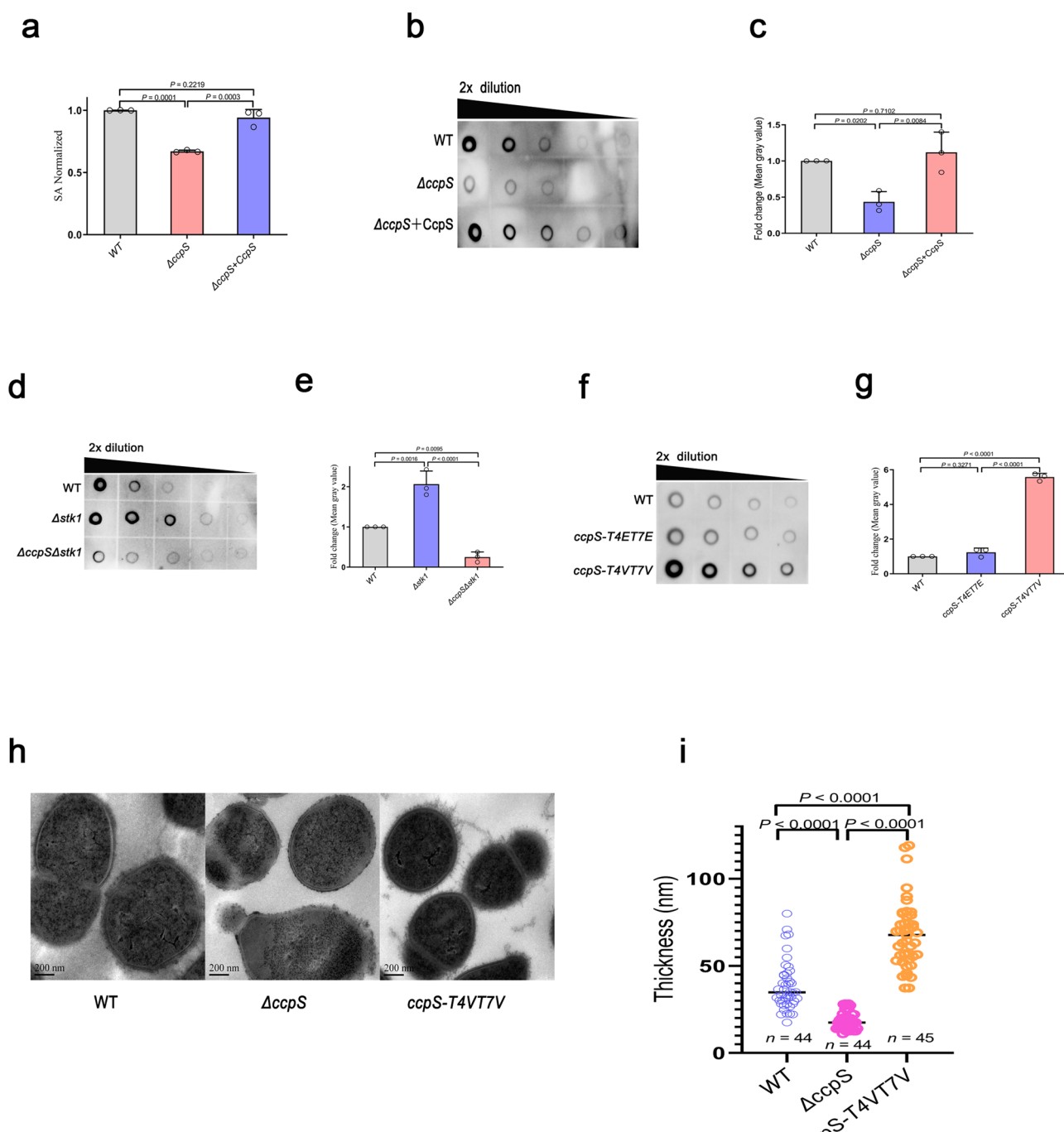

**Fig. 3 | CcpS mediates the capsular polysaccharide synthesis, and the activity of CcpS is tuned by phosphorylation. a** Bar graph showed the cell surface's sialic acid (SA) levels. SA extracted from the cells' surface was quantified by a resorcinol assay. The values were normalized to the WT levels, considered 1. *S. suis* strain ZY05719 (WT), *ΔccpS, ΔccpS* + CcpS (ectopic expression of CcpS by vector pSET2) were grown in THY broth at 37 °C, cultures were normalized to an OD600 = 1.0, and same volume culture medium were assayed. Data represent mean ± SD from *n* = 3 biologically independent experiments. **b**, **d**, **f** Dot blot showed serial dilutions (1:2) of spent growth media spotted on a PVDF membrane and probed with an anti-CPS antibody. **c**, **e**, **g** Bar graph showed the cell-free CPS levels (Density analysis for

**b**, **d**, **f**, respectively, at the second dilution). The values were normalized to the WT levels, considered 1. Data represent mean ± SD from *n* = 3 biologically independent experiments. **h** Transmission electron micrograph of WT (left), *ΔccpS* (middle), and *ccpS-T4VT7V* (right) strains. Scale bar, 200 nm. **i** The average capsule thickness was measured by ImageJ software, *n* indicates the number of cells analyzed, counted 44 cells (WT), 44 cells (*ΔccpS*), and 45 cells (*ccpS-T4VT7V*). Horizontal lines (in black) indicate observation means. Experiments were performed in triplicate. For **a**, **c**, **e**, **g**, **i**, Statistical difference: one-way ANOVA followed by Bonferroni or Tukey's post-tests. *P* values <0.05 indicate significant differences. Source data are provided as a Source Data file.

to gain mechanistic insights into the physiological role of CcpS or Stk1-CcpS system in CPS production, we first screened a bacterial genomic expression library of a *S. suis* laboratory strain using CcpS as a bait and obtained several interaction partners (Supplementary Fig. 4a). Subsequently, bioinformatics analysis showed that three genes *accD* is

responsible for lipid transport and metabolism[48], *nrdR* encodes a transcriptional regulator to control ribonucleotide reductases gene expression[49], and *rsO5395* encodes a putative adhesin, which are functionally unrelated to bacterial capsule synthesis. To test whether the other genes are functionally associated with CPS production, we

constructed gene deletion strains and analyzed cell-associated CPS levels. Among them, tyrosine-protein phosphatase CpsB significantly affected cell-associated CPS levels (Supplementary Fig. 4b). It was reported that CpsB is part of the CpsBCD phosphoregulatory system[11], which is closely related to bacterial CPS production. To further validate the CcpS-CpsB interaction, we performed a GST pull-down assay using the rCpsB-GST and the whole-cell lysates of *S. suis* cells. Western blot analysis showed that CcpS was retained with rCpsB-GST, but not GST (Fig. 4a). Next, to confirm whether CcpS can directly interact with CpsB, a GST pull-down assay using rCcpS-GST and rCpsB-His was performed. The results demonstrated that rCcpS-GST, but not GST, interacted with rCpsB-His directly (Fig. 4b). Furthermore, a bacterial two-hybrid assay indicated that CcpS phosphorylation did not block its interactions with CpsB in *E. coli* (Fig. 4c). In addition, we also performed a similar analysis of the interaction between CcpS and CpsD, which is the cognate tyrosine-protein kinase of CpsB[12]. Interestingly, we found that CcpS also can form an interaction complex with CpsD directly in vivo and in vitro (Supplementary Fig. 4c–e).

Previous studies have evidenced that CpsB and CpsD homologs were involved in CPS synthesis in bacteria[11,12]. *S. suis* also encodes a typical tyrosine kinase/phosphatase system, CpsD/CpsB[9]. However, their exact regulatory roles in *S. suis* capsule synthesis have not been reported yet. To gain insight into the mechanisms regulating CPS synthesis by CpsB and CpsD in *S. suis*, we first applied a bioinformatics approach and searched protein homologs to CpsB/CpsD. Sequence analysis showed that the *S. suis* CpsB protein is evolutionarily conserved and also possesses several conserved residues H136, R139, and R206 (Supplementary Fig. 4f), which may be involved in enzyme catalytic activity[19,21]. Based on CpsD homologs sequence analysis, we speculated that the last three conserved tyrosine residues at the C-terminal tail of CpsD constructing into the conserved tyrosine clusters (Supplementary Fig. 4g), and the residue K49 may be required for its auto-kinase activity[50,51]. Subsequently, to determine whether the *S. suis* CpsD can be phosphorylated in vivo, we analyzed total tyrosine phosphorylation patterns of the whole-cell lysates from the WT strain and *ΔcpsD* mutant strain, the result showed that *S. suis* CpsD tyrosine residues can be phosphorylated and it was the most abundant tyrosine phosphorylation protein in vivo (Supplementary Fig. 4h). Thus, the result also indicated that we can directly detect the tyrosine phosphorylation level of CpsD from whole-cell lysates using a pan anti-phosphotyrosine antibodies. Next, to further test whether the tyrosine phosphorylation occured only at the carboxyl-terminal specific tyrosine clusters of CpsD, we replaced the native *cpsD* locus with *cpsD-3YE* (mimics CpsD-P) or *cpsD-3YF* (mimics non-phosphorylated CpsD), respectively. The resulting *cpsD-3YE* and *cpsD-3YF* mutant strains caused CpsD can not be phosphorylated in vivo (Fig. 4d), while the total level of CpsD protein was unchanged, which suggested that CpsD phosphorylation indeed occurred at C-terminal three tyrosine residues in *S. suis*.

Published studies have demonstrated that CpsC homologs carboxyl-terminal residues (CpsC-ctr) were essential for CpsD kinase activity[50,51]. To determine whether the CpsC-ctr is also required for CpsD kinase activity in *S. suis*, we first performed an auto-phosphorylation assay using recombinant protein rCpsC-ctr and rCpsD in vitro. The results showed that rCpsD phosphorylation can be activated in the presence of rCpsC-ctr (Supplementary Fig. 4i, left two lanes). Interestingly, a chimera protein rCpsC/D, which fused C-terminal 29 amino acids residues of CpsC with CpsD, can be self-activated phosphorylation in *E. coli* (Supplementary Fig. 4i, right two lanes). Our data has shown that CpsB is a manganese ion-dependent phosphatase (Supplementary Fig. 1h). Thus, we first analyzed the effect of CpsB activity on CpsD phosphorylation in *S. suis* by Western blot assay. The results showed that the *ΔcpsB* mutant strain led to the CpsD phosphorylation level of cells increasing significantly compared to the WT strain, while the total CpsD protein levels were unchanged

(Supplementary Fig. 4j, k), which suggested that CpsD phosphorylation level was closely related to CpsB activity.

To further confirm whether the catalytic activity of CpsB directly affected CpsD phosphorylation, we first introduced the rCpsB-R139A and rCpsB-R206A mutations and checked both for catalytic activity with phosphatase assays in vitro. As expected, the substitution of residues R139 and R206 in CpsB for alanine, respectively, resulted in a significant decrease in the catalytic activity of CpsB in vitro (Fig. 4e). Further, we tested the de-phosphorylation of rCpsD-P catalyzed by rCpsB and its variants. The results indicated that WT CpsB can efficiently dephosphorylate CpsD-P in vitro, while the CpsB-R139A and CpsB-R206A variant proteins showed low catalytic activity (Supplementary Fig. 4l, m). These findings further indicated that CpsB plays as an efficient phosphatase to directly remove phosphoryl groups from CpsD-P. For further proof, we carried out a Western blot assay to detect CpsD phosphorylation in WT and mutant strains. The results showed that the CpsD phosphorylation level of cells increased significantly in *cpsB-R139A* and *cpsB-R206A* mutant strains compared to the WT strain (Fig. 4f, g), which is consistent with phosphatase activity assays in vitro (Supplementary Fig. 4l, m). Strikingly, CpsD was not phosphorylated in the *cpsD-K49A* mutant strain, indicating that the residue K49 was required for the auto-kinase activity of CpsD in *S. suis* (Fig. 4f, g).

On the basis of the aforementioned results, we hypothesized that the CcpS may modulate the activity of CpsB or disturb the phosphorylation of CpsD, thereby regulating CPS production. To test this hypothesis, we first quantified the de-phosphorylation of rCpsD-P mediated by rCpsB in the presence of rCcpS or rCcpS-P in vitro. As expected, we observed that non-phosphorylated rCcpS but not phosphorylated rCcpS-P significantly retarded the process of de-phosphorylation of rCpsD-P catalyzed by rCpsB (Fig. 4h, i). Furthermore, we repeated the above experiments using CcpS variants, and found that phospho-ablative rCcpS-T4VT7V also significantly retarded the process of de-phosphorylation of rCpsD-P catalyzed by rCpsB compared to phospho-mimetic rCcpS-T4ET7E (Supplementary Fig. 5a, b). However, rCcpS-P or phospho-mimetic rCcpS-T4ET7E has no significant effect on the process of de-phosphorylation of rCpsD-P catalyzed by rCpsB (Fig. 4h, i and Supplementary Fig. 5a, b). For further proof, we performed biochemical characterization of the activities of rCpsB using pNPP, which is an artificial substrate. The results showed that non-phosphorylated CcpS but not phosphorylated CcpS significantly inhibited the enzyme catalytic activity of CpsB (Supplementary Fig. 5c). Next, to further test whether CcpS can directly disturb the phosphorylation of CpsD, we first incubated rCpsD-P with rCcpS or rCpsB as control, and found that CcpS has no direct effect on CpsD-P (Supplementary Fig. 5d, e). Furthermore, to determine whether the CcpS can influence the auto-kinase activity of CpsD, we first incubated rCpsD with rCcpS variants, after the rCpsC-ctr was added to activate rCpsD phosphorylation. The results showed that the levels of CpsD phosphorylation were similar between rCcpS-T4ET7E and rCcpS-T4VT7V groups (Supplementary Fig. 5f, g). Together, these results indicated that phosphorylation-dependent CcpS can disturb CpsD phosphorylation by modulating CpsB activity.

## Phosphorylation of CcpS modulates the affinity to CpsB and depends on the homo-dimer structure

To gain insights into the architecture and mechanism of CcpS modulates the activity of CpsB, we first solved its structure to 1.6 Å resolution (Supplementary Fig. 7a). The structure of CcpS was solved by molecular replacement and the NMR structure of IreB (PDB code: 5US5) was used as reference[52]. Statistics for X-ray data collection and structure refinements were summarized in Supplementary Table S4. The overall fold of CcpS was similar to IreB[52]. CcpS also formed a homodimer, each monomer containing four α-helices. Helices were connected by a short loop and a single turn of $3_{10}$ helix between helices

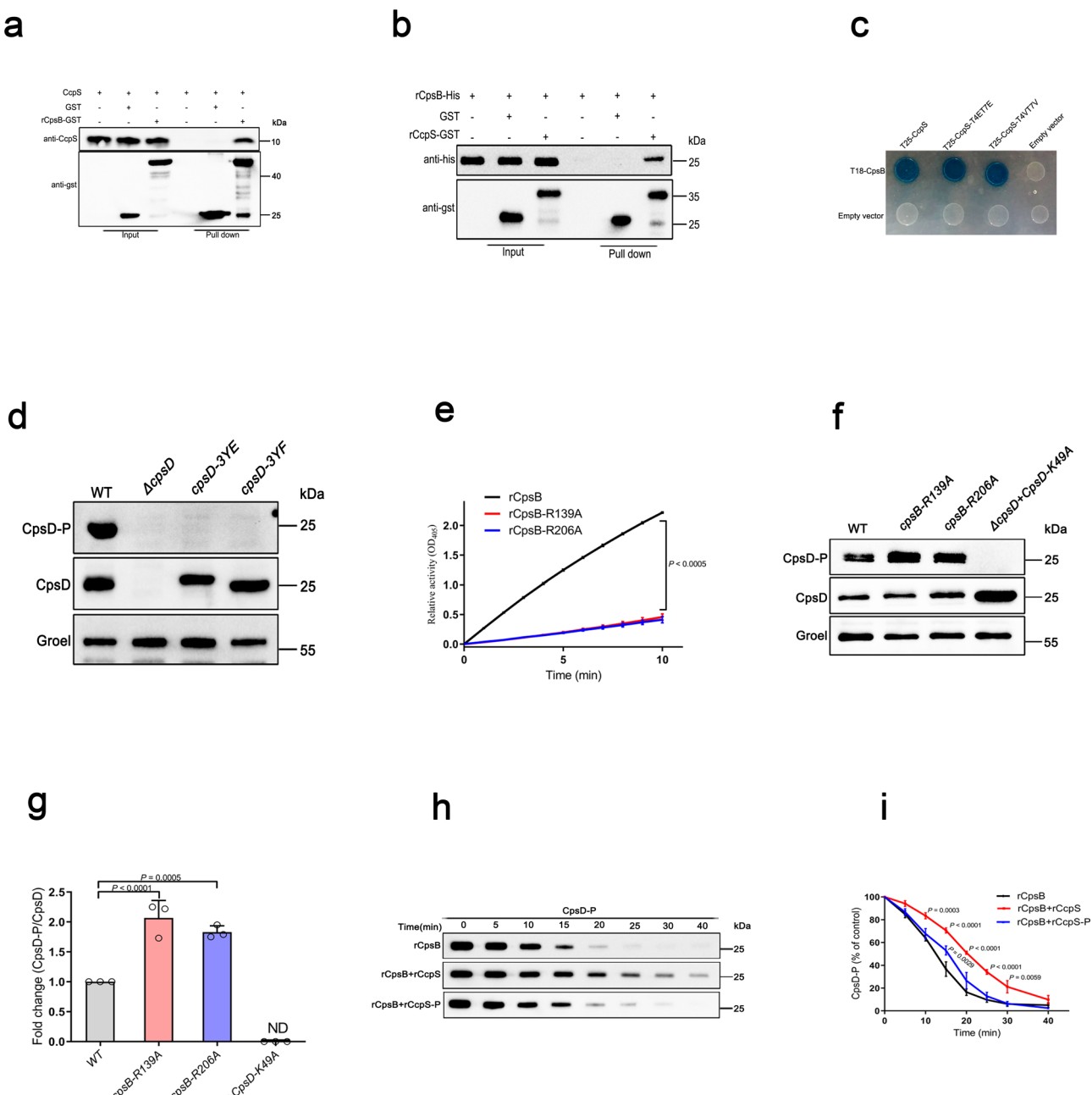

**Fig. 4 | CcpS binds to and modulates CpsB activity. a** Affinity purification of rCpsB-GST from *S. suis* whole-cell lysates pulls down native CcpS and empty GST-tag as control, GST agarose beads capture protein complexes, and the samples were detected with anti-GST and anti-CcpS antibodies, respectively. **b** Pull-down assay confirmed the direct interaction between CcpS and CpsB. Purified rCcpS-GST or empty GST-tag were incubated with rCpsB-His, and protein complexes were captured by GST agarose beads, washed, and eluted in the sample buffer. Fractions were probed with anti-GST and anti-His antibodies. **c** Bacterial two-hybrid assay testing for interactions between CcpS and its variants (CcpS-T4ET7E and CcpS-T4VT7V) with CpsB. Blue colony formation suggests that a direct interaction occurs. **d, f** Anti-phosphotyrosine immunoblot analysis of whole-cell lysates from the indicated strains. Immunoblot shows phosphorylated CpsD and total CpsD protein levels using anti-phosphotyrosine and anti-CpsD antibodies, respectively. Groel: loading control. **e** Biochemical characterization of the activities of CpsB and its variants (CpsB-R139A and CpsB-R206A) using para-nitrophenylphosphate pNPP, which is an artificial substrate. The recombinant rCpsB activity was normalized to Western blot densitometries of CpsB and its variants for each sample, and the WT

CpsB protein activity, was considered 100%. Data represent mean ± SD from $n = 3$ biologically independent experiments. **g** Bar graphs showed the percentage of phosphorylated and total CpsD in different groups (density analysis for **f**). The values were normalized to the WT levels, considered 1. Data represent mean ± SD from $n = 3$ biologically independent experiments. ND non-detected. **h** Phosphatase assays were performed as described in Methods, and the phosphorylated CpsD proteins were detected by SDS gel and Western blot using anti-phosphotyrosine antibodies. Recombinant rCcpS or rCcpS-P were preincubated with rCpsB, respectively, and then adding the rCpsD-P into the mixture to initiate the reaction. Immunoblot showing levels of CpsD phosphorylation using anti-phosphotyrosine antibodies in each group. **i** Bar graphs showed the phosphorylation levels of CpsD in each group at the indicated time point (density analysis for **h**). The values were normalized to the 0 min levels in each group for each time point, considered 100%. Data represent mean ± SD from $n = 3$ biologically independent experiments. Statistical difference: one-way ANOVA (**g**) or two-way ANOVA (**i**) followed by Bonferroni or Tukey's post-tests, compared to only the rCpsB group. $P$ values <0.05 indicate significant differences. Source data are provided as a Source Data file.

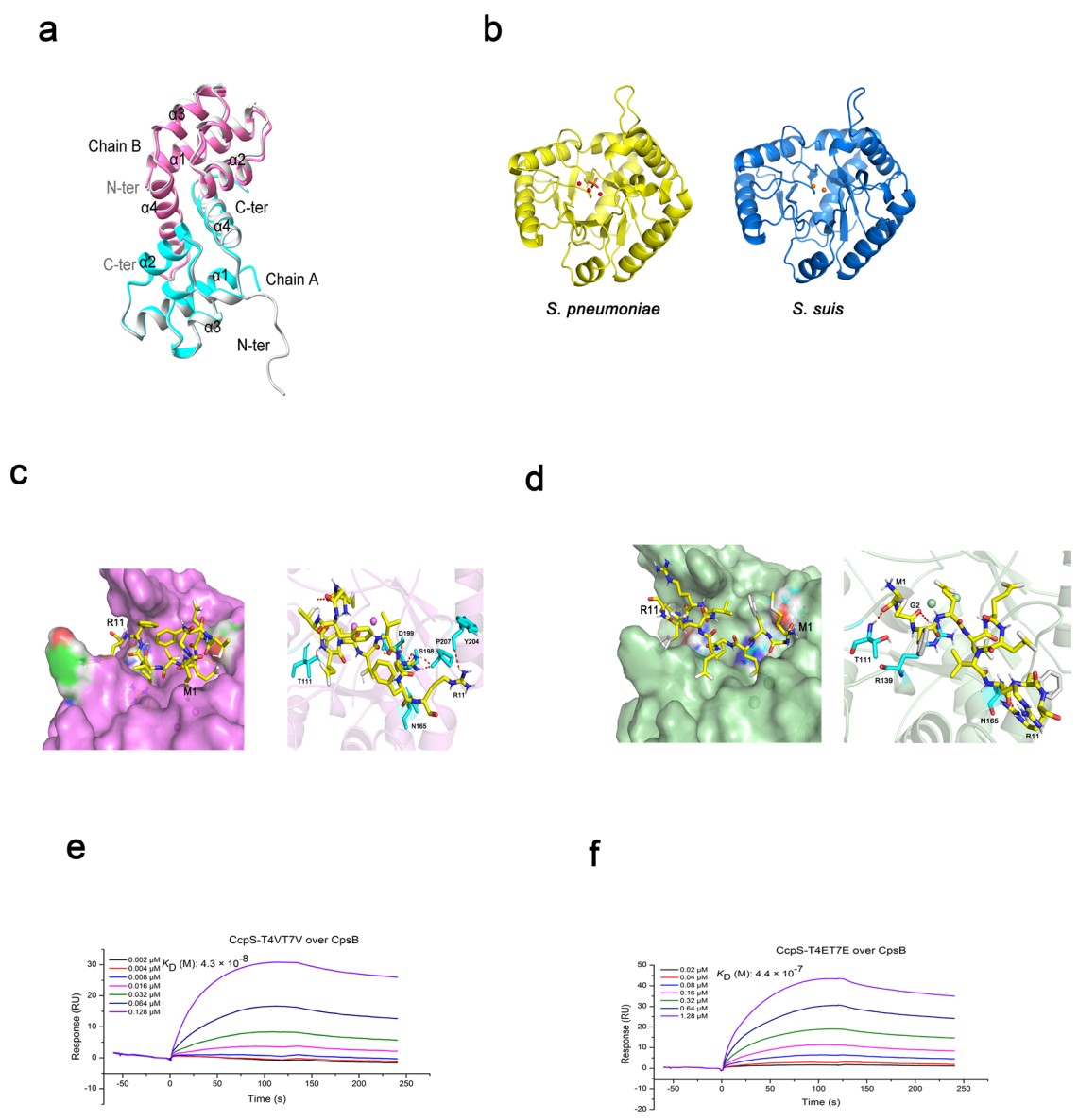

**Fig. 5 | The CcpS affinity to CpsB is tuned by phosphorylation.**
**a** Superimposition of CcpS and its variant CcpS-T4ET7E (gray) X-ray crystal structure, RMSD between 70 pruned atom pairs is 0.398 angstroms. **b** Homology model building of *S. suis* CpsB (right) was performed according to the sequence alignment with the *S. pneumoniae* Cps4B (PDB ID: 2WJF) (left). **c** Surface representation of putative-binding pocket of *S. suis* CpsB with ligand CcpS^N (N means 1–11 amino acids residues at N-terminus of CcpS) shown in yellow sticks (left panel). Detailed interactions of ligand CcpS^N with CpsB and residues critical for ligand binding were shown as cyan sticks and CcpS^N was shown as yellow sticks. The hydrogen bond was shown as red dashed lines (right panel). **d** Surface representation of putative-

binding pocket of CpsB with ligand CcpS-T4ET7E^N (N means 1–11 amino acids residues at N-terminus of CcpS-T4ET7E) shown in yellow sticks (left panel). Detailed interactions of ligand CcpS-T4ET7E^N with CpsB and residues critical for ligand binding were shown as cyan sticks and CcpS-T4ET7E^N was shown as yellow sticks. The hydrogen bond was shown as red dashed lines (right panel). **e, f** SPR analysis of the binding between *S. suis* CpsB and CcpS proteins. Gradient concentrations of the indicated proteins were flowed over immobilized CpsB. Kinetic profiles are shown. CcpS-T4VT7V binding to CpsB (**e**). CcpS-T4ET7E binding to CpsB (**f**). Source data are provided as a Source Data file.

α2 and α3. Structural comparison of CcpS with previously reported CcpS homolog IreB structure (PDB ID: 5US5)[52] revealed that the overall structure was essentially preserved. However, we could not observe the electron density map owing to the highly increased flexibility at the N-terminus of CcpS (Supplementary Fig. 7a). This result was consistent with previous studies[44,52] that CcpS homologs with a highly flexible N-terminus. Intriguingly, our earlier results and previous studies showed that CcpS and its homologs phosphorylation really play important roles in the physiology of bacteria[40,44].

To gain more insights into the phosphorylation events of CcpS, we further analyzed the X-ray crystal structure of CcpS-T4ET7E, which mimics the CcpS-P. Although we obtained a high-resolution X-ray crystal structure of CcpS-T4ET7E, we still could not resolve the

N-terminal flexible region structure. Superimposition of CcpS onto CcpS-T4ET7E folded structure revealed that overall structure was highly similar to each other (Fig. 5a); however, we can observe that a resolvable structure of N-terminal partial region is significantly different between CcpS and its variant CcpS-T4ET7E (Fig. 5a, chain A, N-ter). These results indicated that the highly flexible N-terminal disorder region of CcpS may be not have contributed to intrinsic stability, but was closely related to its binding partners and their functions.

Next, we focused on whether the phosphorylation at the N-terminal disorder region of CcpS affected its binding to CpsB. To this end, we first used the initial 11 residues at the N-terminus of CcpS as ligands. Due to the poorly characterized X-ray crystal structure profile of *S. suis* CpsB, we searched the homologous sequence of CpsB and

found that homologous protein Cps4B (PDB ID: 2WJF)[21] in *S. pneumoniae* was 62% identical in amino acids sequence to *S. suis* CpsB. Then, model building was performed by MODELER in Discovery Studio 3.5 (Fig. 5b). This structure was imported into Autodock Vina to generate docking results with ligands. We observed that ligands could enter into the putative-binding pocket of CpsB (Fig. 5c, left panel). Hydrogen bonds (H bonds) were also found between ligand CcpS[N] and CpsB, and T111, N165, S198, D199, Y204, and P207 in CpsB formed six H bonds with CcpS[N] (Fig. 5c, right panel). Although the ligand CcpS-T4ET7E[N] also docked in the pocket (Fig. 5d, left panel) and T111, R139, and N165 in CpsB formed three H bonds with CcpS-T4ET7E[N] (Fig. 5d, right panel), the binding mode was different from that of ligand CcpS[N] (Fig. 5c, d). Notably, the ligand CcpS-T4ET7E[N] has a lower affinity to CpsB than ligand CcpS[N]. Next, we sought to confirm whether the CcpS phosphorylation that were predicted to mediate the peptide ligands affinity to CpsB in our model. To this end, we tested the binding affinity rCcpS to rCpsB using surface plasmon resonance (SPR). Relative SPR response unit (RU) was induced by CcpS in a dose-dependent manner. As expected, rCcpS-T4VT7V has a higher affinity to rCpsB and much more than rCcpS-T4ET7E at least tenfold (Fig. 5e, f). This result further supported that CcpS phosphorylation indeed modulated the affinity to CpsB. Furthermore, we also tested whether CcpS phosphorylation affected the affinity to CpsD. Interestingly, we found that rCcpS-T4VT7V also showed higher affinity to rCpsD than rCcpS-T4ET7E (Supplementary Fig. 6). In addition, we found that CpsD phosphorylation did not affect the affinity to CcpS (Supplementary Fig. 6a, c or b, d). However, what is the outcome of CpsD interaction with CcpS remains an open question.

Interface residues analysis of CcpS homo-dimer suggested that many amino acids residues were involved in the formation of a dimer (Supplementary Fig. 7a). And the amino acids region I73-L81 among helices α4 was contributed to the hydrophobic core, which was similar to the previously reported structures of CcpS homologs[44,52]. These results indicated that the dimer of CcpS may be important to its functions. To test this hypothesis, we attempted to introduce point mutations and intended to disrupt the dimer. We found that double mutations I73RY80E directly blocked the self-interaction of CcpS (Supplementary Fig. 7b). Furthermore, we found that I73RY80E point mutations also disrupted the interaction between CpsB and CcpS (Supplementary Fig. 7c, d). Importantly, we found that the disruption of CcpS homo-dimer structure directly resulted in its dysfunction in regulating the activity of CpsB (Supplementary Fig. 7e–g) and thus CPS production (Supplementary Fig. 7h). Therefore, we speculated that I73RY80E point mutations might directly destruct 3D structure of CcpS. Together, our data indicated that CcpS phosphorylation modulates the affinity to CpsB, and homo-dimer structure is essential for CcpS biological function.

## CcpS modulates the CpsB activity, thereby altering CpsD phosphorylation and CPS production

To further delineate the mechanisms associated with the activity of CpsB regulated by CcpS, thereby altering CpsD phosphorylation and CPS production in *S. suis*. We first evaluated the consequences of CcpS phosphorylation events on the tyrosine phosphorylation of CpsD. To this end, we performed a Western blot assay to test CpsD phosphorylation. The results showed that *ΔccpS* mutant strain led to a significant decrease in the phosphorylation level of CpsD compared to the WT strain, while the complemented strain was restored to WT strain levels (Supplementary Fig. 8a, b). Notably, the *ccpS-T4VT7V* mutant strain showed a significant increase in the phosphorylation level of CpsD compared to the WT strain, while the *ccpS-T4ET7E* mutant strain showed a similar phosphorylation level of CpsD with the WT strain (Fig. 6a, b). Importantly, CcpS was not phosphorylated in a *Δstk1* mutant strain (Fig. 1e), and the *Δstk1* mutant strain also showed a significant increase in the phosphorylation level of CpsD (Fig. 6c, d).

However, we failed to obtain a single gene *stp1* depletion strain, because the deletion of Stp1 resulted in the simultaneous absence of Stk1 expression in the *S. suis* laboratory strain (Supplementary Fig. 8c, d). Therefore, the double mutant *Δstk1Δstp1* strain with non-phosphorylated CcpS also showed an increase in the phosphorylation level of CpsD (Fig. 6c, d) compared to the WT strain. To further determine whether CcpS modulated CpsD phosphorylation by other pathways besides CpsB in *S. suis*, the effect of CcpS on CpsD phosphorylation was also assessed in the background of the parental *ΔcpsB* mutant strain. We found that *ΔcpsBΔccpS* double mutants strain, *ΔcpsBccpS-T4ET7E* mutant strain, and *ΔcpsBccpS-T4V7V* mutant strain exhibited a similar CpsD phosphorylation level compared to the parental *ΔcpsB* strain (Fig. 6e, f). Furthermore, to investigate whether the difference in phosphorylation level of CpsD is due to changes in total CpsB protein level, we measured the total level of CpsB protein from whole-cell lysates and found no change (Supplementary Fig. 8e, f). Altogether, these results indicated that CcpS affects CpsD phosphorylation levels by modulating CpsB activity, and the activity of CcpS was dependent on the phosphorylation modification by the kinase Stk1.

Subsequently, we further characterized the contribution of CpsD phosphorylation to CPS production in *S. suis*. To this end, we examined the cell-free CPS using an anti-CPS antibody. The results showed that the *cpsD-3YE* mutant strain resulted in a significant increase in the amount of CPS in the medium compared with the WT strain, while the *cpsD-3YF* mutant strain resulted in a significant decrease in the amount of CPS in the medium compared with the WT strain (Supplementary Fig. 8g, h), which suggested that enhanced CpsD phosphorylation promotes the CPS production in *S. suis*. Here, we found that the CPS phenotypes modulated by CpsD phosphorylation were consistent with CcpS phosphorylation events, this provided further support that these factors were indeed involved in the same cell biological process. Furthermore, we compared the cell-associated CPS fluorescence signal (Fig. 6g). We observed that the CPS production of the *ccpS-T4VT7V* mutant strain and *cpsD-3YE* mutant strain were similar and significantly increased compared to the WT strain, while the *ccpS-T4ET7E* mutant strain showed a similar CPS level with WT strain. Moreover, CPS production of the *cpsD-3YF* mutant strain was significantly decreased compared to the WT strain and the non-encapsulated *Δcps* mutant strain no CPS detected (Fig. 6g, h). Altogether, these results indicated that CcpS modulates the CpsB activity, thereby altering CpsD phosphorylation, which is directly associated with CPS production, and thus ultimately mediates the CPS production in *S. suis*.

## CcpS phosphorylation constitutes a widespread mechanism of signal transduction and regulation in bacteria

Interestingly, our bioinformatics data showed that most Firmicutes encode a conserved CcpS homolog, which harbors a highly conserved threonine residue Thr7 (Supplementary Fig. 9a, b). Furthermore, multiple sequence alignment analysis showed that CcpS homologs possess the conserved secondary structure, including four compact α-helices, and has multiple conserved residues (Supplementary Fig. 9c). Then, to further tested whether these CcpS homologs phosphorylation events are conserved, we reconstituted the Stk1-CcpS homologs system in *E. coli*. As expected, all the CcpS homologs can be phosphorylated by Stk1 kinase from *S. suis* (Fig. 7a). Moreover, bacterial two-hybrid assays in *E. coli* indicated that all CcpS homologs self-interacted (Fig. 7b). Interestingly, we also found that all CcpS homologs and CpsB from *S. suis* can form an interaction complex in a heterologous host organism (Supplementary Fig. 9d). These results further supported that CcpS is a highly conserved protein in almost the whole Firmicutes, and thus the CcpS homologs phosphorylation may be a widespread signaling mechanism in these bacteria.

*S. agalactiae*, the leading cause of bacterial sepsis and meningitis in human neonates[53], is also covered with a thick capsule layer. To further understand the function of the CcpS homolog in the CPS

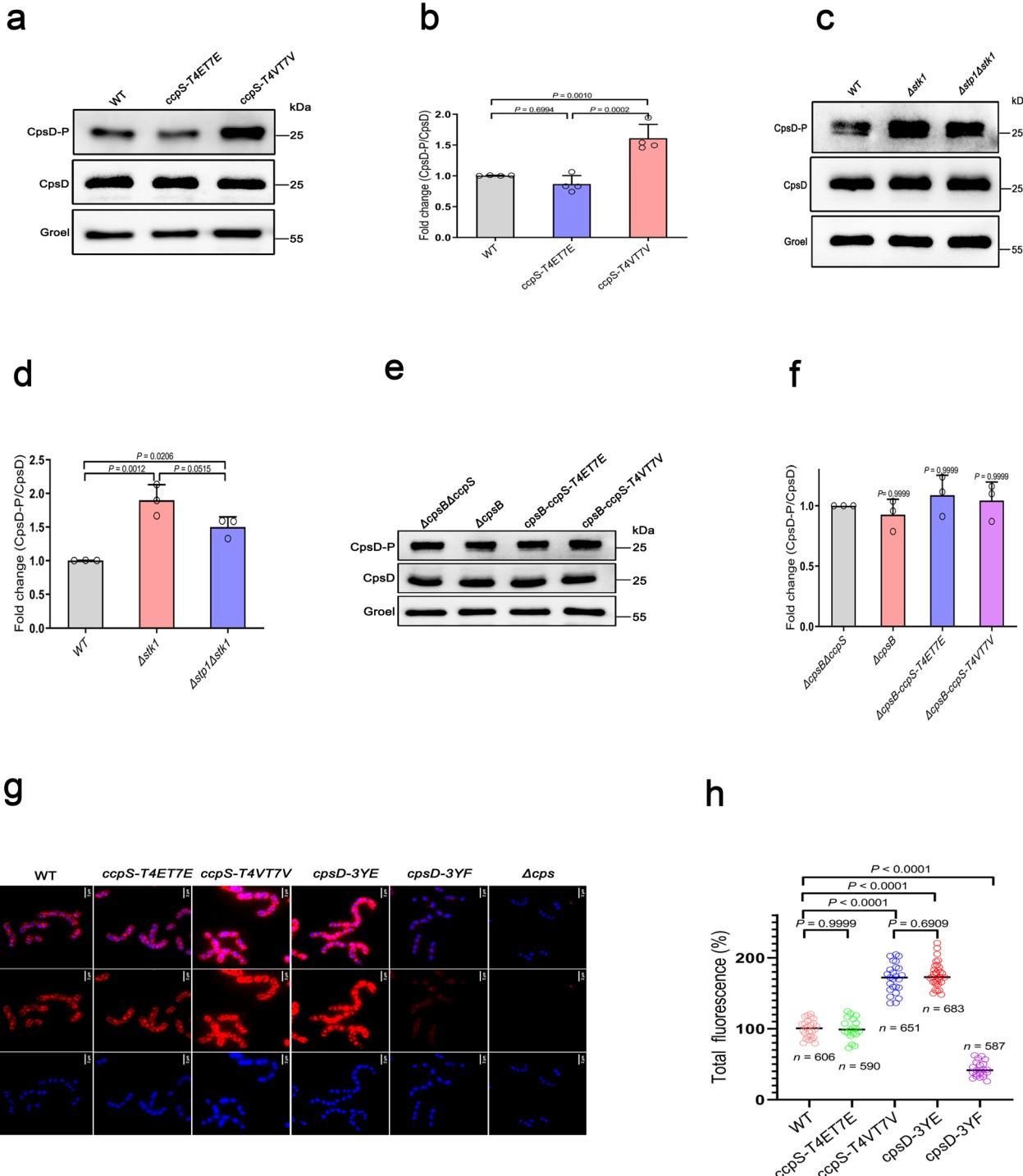

**Fig. 6 | CcpS regulates CpsD phosphorylation and then influences CPS synthesis in *S. suis*. a, c, e** Anti-phosphotyrosine immunoblot analysis of whole-cell lysates from the indicated strains. Immunoblot shows phosphorylated CpsD and total CpsD protein levels using anti-phosphotyrosine and anti-CpsD antibodies, respectively. Groel: loading control. **b, d, f** Bar graphs showed the percentage of phosphorylated and total CpsD in different groups (density analysis for **a, c, e**, respectively). The values were normalized to the WT levels, considered 1. Data represent mean ± SD from $n = 4$ (**b**) and $n = 3$ (**d, f**) biologically independent experiments. **g** Detection of CPS in WT and mutant strains. CPS were immunode-tected with a mouse anti-serotype 2 CPS polyclonal antibody. CPS fluorescent signal (red, middle row), DAPI (blue, bottom row), and overlays between DAPI and CPS fluorescence images (upper row) are shown. Scale bar, 2 μm. **h** Quantification of the CPS fluorescent signal in WT and mutants strains observed in (**g**), the values were normalized to the WT levels, considered 100%. $n$ indicates the number of cells were analyzed, counted 606 cells from 26 images (WT), 590 cells from 19 images (*ccpS-T4ET7E*), 651 cells from 25 images (*ccpS-T4VT7V*), 683 cells from 32 images (*cpsD-3YE*), and 587 cells from 22 images (*cpsD-3YF*). Horizontal lines (in black) indicate observation means. Experiments were performed in triplicate. For **b, d, f, h**, Statistical difference: one-way ANOVA followed by Bonferroni or Tukey's post-tests. $P$ values <0.05 indicate significant differences. Source data are provided as a Source Data file.

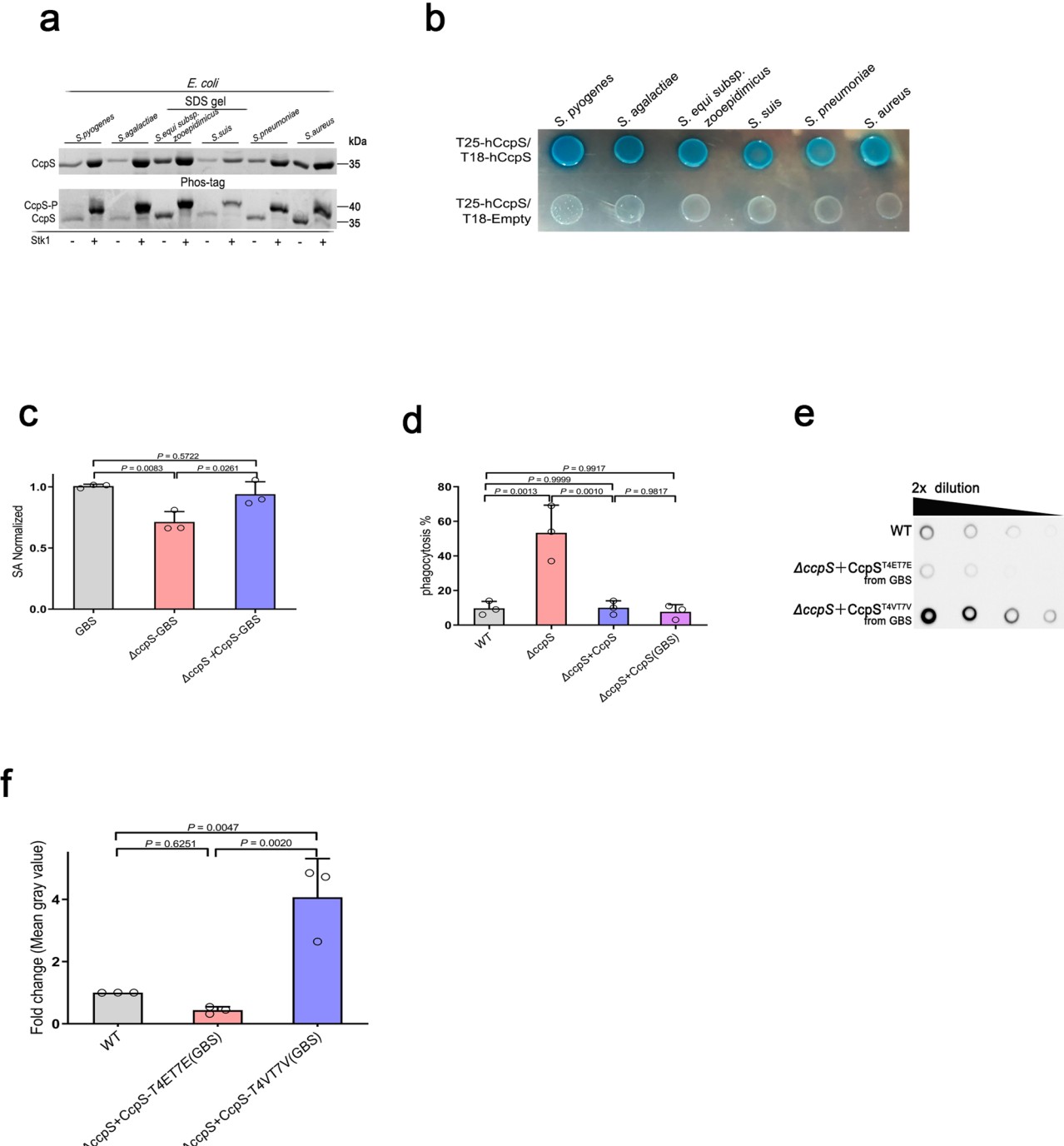

**Fig. 7 | CcpS phosphorylation-modulation mode constitutes a widespread mechanism of signal transduction and regulation in Gram-positive bacteria.**
**a** The homologous CcpS-GST proteins were expressed in *E. coli* in the absence or presence of *S. suis* Stk1, GST agarose beads were used to pull down homologous CcpS-GST and the samples were loaded onto Phos-tag gel to analyze the phosphorylation of CcpS homologs, samples were also loaded onto a standard SDS gel and Coomassie stain. For this experiment, CcpS and its homologs from *Streptococcus pyogenes, Streptococcus agalactiae, Streptococcus equi subsp. zooepidemicus, S. suis, Streptococcus pneumoniae,* and *Staphylococcus aureus* were analyzed.
**b** Bacterial two-hybrid assay testing for self-interactions of CcpS homologs. Blue colony formation suggests that a direct interaction occurs. **c** Bar graph showed the cell surface's sialic acid (SA) levels of *S. agalactiae* strain A909 (WT) and its derivative strains. SA extracted from the cells' surface was quantified by a resorcinol assay. The values were normalized to the WT levels, considered 1. Data represent mean ± SD from *n* = 3 biologically independent experiments. **d** Bacterial

phagocytosis as described in Methods, the results are expressed as the percentage of CFU recovered bacteria/initial bacterial CFU. For this experiment, *S. suis* strain ZY05719 (WT), *ΔccpS, ΔccpS* + CcpS (ectopic expression of CcpS from ZY05719 by vector pSET2), and *ΔccpS* + CcpS (GBS) (ectopic expression of CcpS from A909 by vector pSET2) were analyzed. Data represent mean ± SD from *n* = 3 biologically independent experiments. **e** Dot blot showing serial dilutions (1:2) of spent growth media spotted on a PVDF membrane and probed with an anti-CPS antibody. For this experiment, *S. suis* strain ZY05719 (WT), *ΔccpS*+CcpS-T4ET7E (ectopic expression of CcpS-T4ET7E from A909 by vector pSET2), and *ΔccpS*+CcpS-T4VT7V (ectopic expression of CcpS-T4VT7V from A909 by vector pSET2) were analyzed. **f** Bar graph showed the cell-free CPS levels (Density analysis for **e**, at the second dilution). The values were normalized to the WT levels, considered 1. Data represent mean ± SD from *n* = 3 biologically independent experiments. For **c, d, f**, Statistical difference: one-way ANOVA followed by Bonferroni or Tukey's post-tests. *P* values <0.05 indicate significant differences. Source data are provided as a Source Data file.

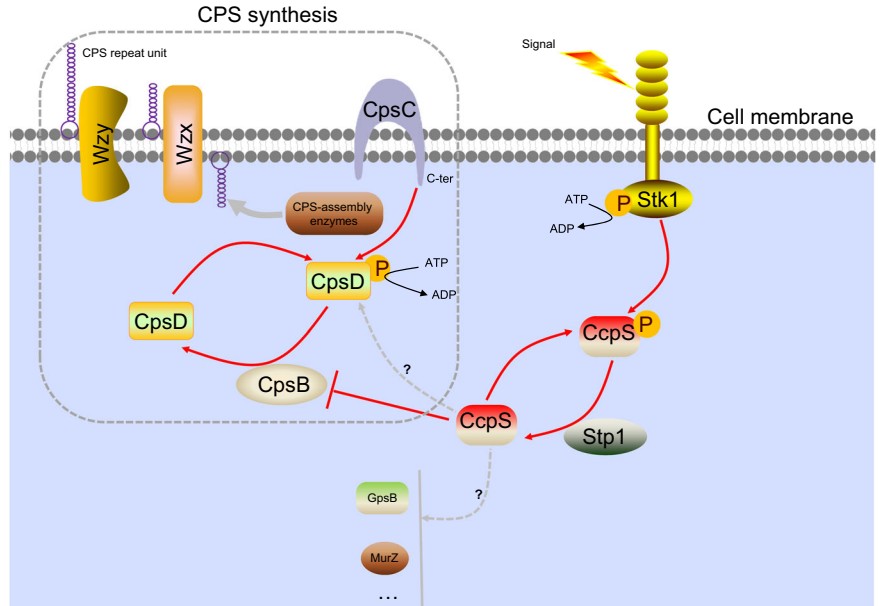

**Fig. 8 | A model schematic depicting the mechanism of CPS synthesis coordination by CcpS phosphorylation.** In *S. suis*, CcpS is specifically phosphorylated by the Stk1/Stp1 system. Non-phosphorylated CcpS inhibits CpsB activity. CpsB then modulates the phosphorylation levels of CpsD. CpsB and CpsD are the important components of the CpsBCD phosphoregulatory system in the Wzx-Wzy pathway, which is responsible for CPS synthesis in *S. suis*. The dysfunction of CpsB and CpsD phosphorylation directly cause abnormal CPS production. CPS-assembly enzymes: These include initiating phosphoglycosyl transferase and other late-stage enzymes. Source data are provided as a Source Data file.

synthesis of *S. agalactiae*, we attempted to construct *ccpS* homolog deletion in *S. agalactiae*. Interestingly, we found that Δ*ccpS* (GBS) mutant strain also significantly decreased the cell-associated CPS production compared to the WT strain in *S. agalactiae* (Fig. 7c), while the amount of cell-associated CPS by the complemented strain was restored to WT strain levels. Furthermore, the Δ*ccpS* (GBS) mutant strain also led to a decrease in the amount of CPS in the medium compared with the WT strain and complemented strain (Supplementary Fig. 9e, f). In addition, the bacterial two-hybrid assays showed that *S. agalactiae* proteins CcpS and CpsB also formed an interaction complex in *E. coli* (Supplementary Fig. 9g). Subsequently, to further explore whether CcpS homologs functions were evolutionarily conserved, we ectopically expressed CcpS homolog, which from *S. agalactiae*, in the heterologous host *S. suis* Δ*ccpS* mutant strain, and the resulting strain can recover the ability to resist phagocytosis of macrophages to WT strain level (Fig. 7d). Moreover, *S. suis* Δ*ccpS* mutant strain ectopically expressed CcpS-T4ET7E and CcpS-T4VT7V homologs, which from *S. agalactiae*, and the resulting strain Δ*ccpS* + CcpS-T4VT7V (GBS) showed a significant increase in the amount of CPS in the medium compared with the WT strain, while strain Δ*ccpS* + CcpS-T4ET7E (GBS) was similar with WT strain (Fig. 7e, f). In addition, we observed that CcpS deletion reduce virulence in animal models (Supplementary Fig. 9h). Overall, these results and the existence of CcpS homologs in many Firmicutes indicated that possible widespread signaling and regulation in these Gram-positive bacteria.

## Discussion

The bacterial CPS has been postulated to be involved in stress adaptation, pathogenesis, and long-persistence[1,2]. To date, how bacteria sense extracellular signals to regulate CPS production is not fully understood. Here, we present evidence that bacteria have evolved sophisticated mechanisms for building a linkage between signal transduction by Stk1/Stp1 system and the Wzx-Wzy pathway for CPS synthesis. In our model, CcpS acts as a cornerstone (Fig. 8), not only transfers the signals from Stk1 sensing extracellular stimuli, but also transforms the signals into specific physiological functions, here, to regulate the enzyme catalytic activity of CpsB, which plays an important role in Wzx-Wzy pathway[2]. Subsequently, CpsB can directly regulate the phosphorylation level of CpsD in *S. suis*, and ultimately the dysfunction of CpsB and CpsD phosphorylation directly lead to abnormal CPS production in *S. suis*. These findings reveal an unexpected molecular link between two important biological processes: the ability of bacteria to rapidly respond to changes around them by the Stk1-CcpS axis and to modulate CPS production by the Wzx-Wzy pathway.

The Stk1-Stp1 pair is conserved in Gram-positive bacteria, which has fueled interest in elucidating their potential roles in bacterial signaling and regulation[24,25]. CcpS, as a new direct substrate of the Stk1/Stp1 system, presented different phosphorylation patterns in *S. suis* pathogen when it was against various stressors, which further supported that the Stk1-Stp1 pair plays an important role in bacterial biological signaling and regulation. Interestingly, the CcpS phosphorylation level of cells dramatically decreased in several extracellular stimuli, such as an acidic environment, oxidative stress, and starvation (Fig. 2), while the CcpS phosphorylation level of cells significantly increased in a nutrient medium, thus suggesting that the Stk1-CcpS axis may be an important signaling pathway that helps bacteria to survive against stressors or different host niches. Notably, a previous study also demonstrated that CcpS homolog IreB phosphorylation was active when cells were against various stressors[43]. Further studies will be required to investigate the effect of these biological processes on bacterial virulence in different host niches. The previous study showed that eSTK could sense PG fragments and induce germination[26], suggesting bacterial eSTK indeed can sense extracellular signals. However, if the above extracellular stimuli can stimulate Stk1 activity directly or indirectly, while the mechanism of which stimuli is sensed by cells and how it regulates Stk1 kinase activity remains to be elucidated. eSTKs are the new hallmark of bacterial phosphosignaling and it possess multiple substrates and controls all aspects of bacterial physiology[24,25]. Here, we show that the Stk1-CcpS axis is active (Fig. 2), indicating their potential biological signaling and regulation functions in the cell physiology of *S. suis*.

Interestingly, previous studies suggested that CcpS homologs IreB/ReoM were associated with cephalosporin resistance in *E. faecalis*[40] and

*L. monocytogenes*[44], while the effect of CcpS and its phosphorylation on cephalosporin resistance were negligible in *S. suis* (Supplementary Fig. 3b). In addition, CcpS homolog ReoM whose function has recently been described in *L. monocytogenes* that ReoM phosphorylation controlled ClpCP-dependent proteolytic degradation of MurA, which catalyzes the first committed step in PG biosynthesis[44], and ultimately affect the ceftriaxone resistance. We found that *S. suis* also encode a homolog MurA1 and a second paralogue MurZ. Importantly, a single Δ*murA1* or Δ*murZ* mutant strain was easily obtained and Δ*murA1* or Δ*murZ* mutant strain has no effect on *S. suis* resistance to cephalosporin, while double *murA1- murZ* deletion was lethal to *S. suis*, which indicated that MurZ and MurA1 may be functionally replaced in *S. suis*. Previous studies also showed that MurZ and MurA homologs can be functionally replaced in *S. aureus* and *S. pneumoniae*[54,55], so that proteolytic degradation of the primary enzyme can be tolerated. Furthermore, both the *S. suis* WT strain and CcpS mutants strain exhibited similar MurA1 and MurZ protein levels (Supplementary Fig. 3d–g), suggesting that the functions of CcpS phosphorylation proposed were not mediated by regulated proteolysis. Therefore, CcpS must have other functions due to the Stk1-CcpS axis is active in *S. suis*.

Surprisingly, this work defines a direct connection between Stk1 kinase activity and CpsB function mediated by CcpS (Fig. 8). We found that the Stk1-CcpS system was associated with resist phagocytosis of macrophages and was required for CPS production in *S. suis* (Supplementary Fig. 3k, l and Fig. 3). Interestingly, CPS is an important virulence factor that can help bacteria to resist phagocytosis[1], and CPS is required for *S. suis* resisting phagocytosis of macrophages (Supplementary Fig. 3k)[56]. Thus, we speculate that the Stk1-CcpS system modulates CPS production, thereby contributing to *S. suis* resisting the phagocytosis of macrophages and regulating bacterial virulence. However, protein sequence analysis and functional domain prediction revealed that CcpS may indirectly participate in CPS synthesis in *S. suis*. Notably, one of the interaction partners of CcpS, tyrosine-protein phosphatase CpsB (Supplementary Fig. 4a), which is part of the CpsBCD phosphoregulatory system belonging to the Wzx-Wzy pathway[2]. Interestingly, non-phosphorylated CcpS could block the de-phosphorylation process of CpsD-P catalyzed by CpsB in vitro and enhance the phosphorylation level of CpsD in vivo. Further, enhanced CpsD phosphorylation led to an increase in CPS production of *S. suis*. However, phosphorylated CcpS has no significant effect on CpsB activity and did not disturb the de-phosphorylation process of CpsD-P catalyzed by CpsB in vitro (Fig. 4h, i and Supplementary Fig. 5a, b), while this seems to be inconsistent with in vivo data. Because CpsD phosphorylation levels in the phospho-mimetic *ccpS-T4ET7E* mutant strain was similar to the WT strain (Fig. 6a, b), however, Δ*ccpS* mutant strain led to a significant decrease in the phosphorylation level of CpsD compared to WT strain (Supplementary Fig. 8a, b). The most likely reason for this is that the glutamic acid substitution can not fully mimicked phosphorylation at Thr4 and Thr7 in vivo. Instead, CcpS-T4ET7E possessed properties of both the phosphorylated and non-phosphorylated forms in vivo. Notably, it is a universal phenomenon that aspartate or glutamic acid are unfaithful chemical mimics of phosphothreonine[57,58], including CcpS homologs IreB in *E. faecalis*[40] and ReoM in *L. monocytogenes*[44].

The Δ*stk1* mutant strain with non-phosphorylated CcpS also led to a significant increase in CpsD phosphorylation and CPS production, which are consistent with previous studies[36,59], in which CpsD homologs phosphorylation level were strongly enhanced when the Stk1 homologs were depleted in vivo. Interestingly, it has reported that Stk1 homolog PknB modulates the activity of tyrosine kinase complex CapA1B1$_{fus}$ by directly phosphorylating CapB1[36], which seems different from our model. It could be speculated that, on the one hand, due to the specificity of the strain. Although the tyrosine cluster at the C-terminus of CpsD is conserved also other important residues (Supplementary Fig. 4g), the whole sequence of CpsD homologs in *S. suis*

and *S. aureus* are quite different. To the best of our knowledge, the cross phosphorylation of CpsD homologs by Stk1 homologs have not been exactly reported in Streptococci yet. Another possibility is that other factors were involved in enhanced CpsD homolog phosphorylation level in a Δ*pknB* mutant strain, because an exchange of Thr8 to Ala did not affect CapA1B1$_{fus}$ auto-phosphorylation level in vivo [37], while CPS production was substantially elevated in a Δ*pknB* mutant strain of *S. aureus* as well as in Δ*stk1* mutant strain of *S. suis*. Thus, other potential factors downstream of Stk1 homologs may be involved in bacterial CPS synthesis and need further investigation.

Given that many proteins involved in the pathogenesis of human diseases contain intrinsically disordered regions (IDRs)[60], suggesting these proteins with IDRs play an important role in cell physiology. Interestingly, the data presented here provide new insight into the molecular mechanism of a protein with functional disorder region controls bacterial physiology. We show that CcpS and CcpS-T4ET7E variant folded structures were highly matched (Fig. 5a). Although the electron density map of the N-terminus was not clear, the partially folded structure of the N-terminus between CcpS and CcpS-T4ET7E chain A crystal structure was obviously different (Fig. 5a, chain A, N-ter). Although the exact mechanism by which non-phosphorylated CcpS inhibits CpsB activity is still unknown, analysis of molecular docking and affinity showed that non-phosphorylated CcpS has a higher affinity to CpsB much more than CcpS-P (Fig. 5c–f). It has been reported that non-phosphorylated EIIA$^{Glc}$ can inhibit the activity of various enzymes through direct interaction[61], EIIA$^{Glc}$ binding to domain II of ecGK and modulates the glycerol phosphorylation activity via long-range conformational changes[62]. Another interesting study reported that LsrK activity is inhibited when bound to non-phosphorylated HPr, revealing new linkages between QS activity and sugar metabolism[63]. In addition, phosphorylation-dependent CcpS directly formed an interaction complex with CpsB and modulated its activity. Therefore, CcpS may play a similar role to EIIA$^{Glc}$ for ecGK or HPr for LsrK, in which non-phosphorylated CcpS has a higher affinity to CpsB and inhibits CpsB activity much more than the CcpS-P. Intrinsically disordered protein regions (IDPRs) of hybrid proteins possessing both structured and disordered domains, which adopt an ensemble of conformations in solution, yet they are functional[64]. In fact, IDPRs frequently serve as major regulators of their binding partners, are controlled by extensive post-translational modifications[60,64]. CcpS as a typical IDPRs, which contained functional phosphorylation sites at the N-terminal disorder region. In addition, IDPRs may be involved in signaling interactions where they undergo constant "bound–unbound" transitions, thus acting as dynamic and sensitive "on-off" switches[60]. Importantly, CcpS was functionally dependent on IDRs, which was also fine-tuned by phosphorylation, and the homo-dimers structure was essential for its function. Like its homologs IreB/ReoM, which function as dimers[44,52]. Therefore, it is possible that CcpS forms an obligate dimer and undergoes constant "non-phosphorylated-phosphorylated" transitions in physiological processes, thus acting as an "on-off" switch to modulate CpsB activity.

Given the high participation of IDPRs in protein-protein interactions[60], it is reasonable to speculate that CcpS possessed various interaction partners. As expected, CcpS has a variety of functionally irrelevant interaction partner proteins (Supplementary Fig. 4a). Interestingly, recent studies suggested that a cell cycle regulator, GpsB, functions as an adapter protein that mediates the interaction between different partners to generate larger protein complexes at specific sites in a bacterial cell cycle-dependent manner[33]. Thus, it remains an open question whether CcpS similarly integrates multiple interaction partners to coordinate the biological process of cells. Notably, interaction analysis suggests that GpsB is the partner of CcpS, while the deletion of GpsB has no effect on CPS production in *S. suis* (Supplementary Fig. 4b). Future experiments will be aimed at establishing whether CcpS and GpsB function as a joint linking the parallel regulatory pathways.

The Wzx-Wzy pathway is the most widespread capsule synthesis system in bacteria[2], in which the phosphoregulatory system CpsBCD plays a key role in regulating CPS production. The data presented here demonstrated that CpsBCD phosphoregulatory system was also important to CPS productions in *S. suis*. Interestingly, both CpsB and CpsD were the interaction partners of CcpS. Although CcpS phosphorylation also modulated the affinity to CpsD, the outcome of CpsD interaction with CcpS was unknown. Notably, previous studies have reported that tyrosine kinase CpsD homologs can mediate the phosphorylation of endogenous substrates[65,66]. Thus, whether CcpS modulates the process of endogenous substrates phosphorylated by CpsD needs further investigation. The results presented here suggested that CcpS was involved in the biological process of CPS synthesis by regulating CpsB activity is conserved (Fig. 7 and Supplementary Fig. 9e–g), which provided a previously unknown linkage between Stk1/Stp1 system and Wzx-Wzy pathway at least in some gram-positive bacteria. Because not all gram-positive bacteria with CPS are generated by the Wzx-Wzy pathway, such as *S. pyogenes* with hyaluronic acid capsule[67]. Furthermore, CcpS was evolutionarily conserved in Firmicutes and with a unique structure (Fig. 7a, b and Supplementary Fig. 9a–d). Thus, the highly conserved phosphorylation site Thr7 of CcpS is associated with conserved STK kinase, indicating the existence of a possible widespread signaling function and similar regulation mode in other bacteria.

In conclusion, our findings reveal a previously unknown mechanism, that a single protein ties Stk1/Stp1 system and Wzx-Wzy pathway together in bacteria. In this respect, to further explore the molecular mechanisms of CcpS phosphorylation in modulating CPS production when bacteria against different host niches will provide useful insights into bacterial pathogenesis. Also, the signal transduction and regulatory mode are widely distributed in bacteria due to the unique structure of CcpS and its phosphorylation are conserved. Therefore, the mechanism of the CcpS function proposed here would pave the way to reveal additional modules involving CcpS or its homologs with partner interaction, which will deepen our understanding of cell biological regulatory networks in bacterial pathogens.

## Methods

### Strains and growth conditions

All *S. suis* and its derivatives in the study were grown at 37 °C in Todd Hewitt broth (THB) or THY medium (THB containing 2% Yeast extract) at pH 7.4 unless otherwise stated. *Escherichia coli* were grown in an LB medium (Oxoid). *E. coli* DH5α strain was used for cloning. *E. coli* BL21 (DE3) was used for protein overexpression. All strains used list in Supplementary Table S1. Growth was monitored by optical density (OD) reading at OD 600 nm for *S. suis* or *E. coli*. Growth curve measurement: Overnight cultured indicated bacteria were diluted $OD_{600}$ to 0.01–0.02, the $OD_{600}$ were read automatically every 15 min for 8–10 h. The following concentrations of antibiotics were used: 50 µg ml$^{-1}$ kanamycin, 100 µg ml$^{-1}$ ampicillin, 100 µg ml$^{-1}$ streptomycin, spectinomycin: 50 µg ml$^{-1}$ (*E. coli*), 100 µg ml$^{-1}$ (*S. suis*) unless otherwise noted.

### Strains construction, plasmids, and primers

The strain *E. coli* DH5α was used for standard cloning. The WT strain of *S. suis* used was the clinical isolate ZY05719 (GI: 820722437), and all the mutants were derived from this strain unless otherwise stated. *S. suis* gene deletion or point mutants were constructed using the standard allele exchange method with different derivatives of thermosensitive suicide vector pSET4s as previously described procedure in refs. 39,68. Each construct was confirmed by PCR and Sanger sequencing. The ectopic expression of the bacterial genes by plasmid, as previously described in ref. 69. All plasmids and oligonucleotide primers used in this study were summarized in Supplementary Tables S2, 3.

### Construction of plasmids

**Plasmid plS1-9**. To construct plasmid plS1, the upstream and downstream flanking regions of gene *rs02080* were amplified from *S. suis* using the primers ol1/ol2 and ol3/ol4, respectively. The resulting two fragments were fused by the third PCR reaction using the primers ol1/ol4, and the resulting third fragment was inserted into the corresponding site of pSET4s digested with EcoRI and BamHI. Likewise, the upstream and downstream flanking regions of genes *rs00400*, *rs02785, rs02780, rs02020, rs07665, rs08535, rs02790*, and *rs02795* were amplified from *S. suis* using the primers ol5-ol36, respectively, and construct corresponding plasmids plS2-plS9.

**Plasmid plS10-plS12 and plS19**. The gene *rs00400* with its upstream and downstream flanking region was amplified from *S. suis* using the primers ol37/ol38. The DNA fragment was inserted into the corresponding site of pSET4s digested with EcoRI and BamHI, resulting in the plasmid plS10. To obtain the point mutation, the whole plasmid sequence was amplified using the primers ol39/ol40, ol41/ol42, ol55/ol56 with plasmid plS10 as the template, resulting products were digested by DpnI and construct corresponding plasmid *plS10-plS12* and *plS19*.

**Plasmid plS20 and plS22**. To construct plasmid plS20, the gene *rs02785* with its upstream and downstream flanking region was amplified from *S. suis* using the primers ol57/ol58. The resulting fragment was inserted into the corresponding site of pSET4s digested with EcoRI and BamHI. The whole plasmid sequence was amplified using the primers ol59/ol60 and ol61/ol62 with plasmid plS10 as the template for its point mutation. The resulting products were digested by DpnI and constructed corresponding plasmid plS21 and plS22.

**Plasmid plS23-plS26**. The gene *rs02780* and its upstream and downstream flanking regions from *S. suis* were amplified by primer ol63/ol64, and the resulting fragment was inserted into the corresponding site of pSET4s digested with EcoRI and BamHI to construct plasmid plS23. To construct RS02780 point mutations, the whole plasmid sequence was amplified using the primers ol65/ol66, ol67/ol68 with plasmid plS23 as the template digested by DpnI and constructed corresponding plasmid plS24, plS25. The upstream promoter region of *cps* locus and gene *rs02785* from *S. suis* were amplified using the primers ol77/ol78 and ol79/ol80, respectively, The resulting two fragments were fused by the third PCR reaction using the primers ol77/ol80, and the resulting third fragment was inserted into the corresponding site of pSET2 digested with EcoRI and BamHI. and corresponding plasmids were constructed plS26.

**Plasmid plS27**. The gene *rs00400* with its promoter region from *S. suis* was amplified using primers ol71/ol72. The resulting fragment was inserted into the corresponding site of pSET2 digested by EcoRI and BamHI and constructed the plasmid plS27.

**Plasmid plS28 and Plasmid plS29**. To construct plasmid plS28, the upstream promoter region of *cps* locus and gene *rs02780* from *S. suis* were amplified using primers ol73/ol74 and ol75/ol76, respectively. The resulting two fragments were fused by the third PCR reaction using the primers ol73/ol76, and the resulting third fragment was inserted into the corresponding site of pSET2 digested with EcoRI and BamHI. To construct RS02785 point mutations, the whole plasmid sequence was amplified using the primers ol69/ol70 with plasmid plS26 as the template digested by DpnI and constructed corresponding plasmid plS29.

**Plasmid plS30-plS33**. To construct plasmid plS30, the gene *rs00400* from *S. suis* was amplified using the primers ol81/ol82. The resulting fragment was inserted into the corresponding site of pET-28a, digested

by EcoRI and BamHI. To construct RS00400 point mutations, the whole plasmid sequence was amplified using the primers ol83/ol84, ol85/ol86, and ol87/ol88 with plasmid plS30 as the template, and the resulting products were digested by DpnI and constructed corresponding plasmid plS31, plS32, and plS33 respectively.

**Plasmid plS34-plS36**. The gene *rs02785* from *S. suis* was amplified using the primers ol89/ol90. The resulting fragment was inserted into the corresponding site of pET-28a, digested by EcoRI and BamHI, and constructed plasmid plS34. To construct the RS02785 point mutations, the whole plasmid sequence was amplified using the primers ol91/ol92 and ol93/ol94 with plasmid plS34 as the template. The resulting products were digested by DpnI and constructed corresponding plasmid plS35 and plS36, respectively.

**Plasmid plS37-plS40**. To construct plasmid plS37, the gene *rs02780* from *S. suis* was amplified using ol95/ol96. The resulting fragment was inserted into the corresponding site of pET-28a digested by EcoRI and BamHI. To construct the RS02780 point mutations, the whole plasmid sequence was amplified using the primers ol97/ol98, ol99/ol100, and ol101/0l102 with plasmid plS37 as the template, and the resulting products were digested by DpnI and constructed corresponding plasmid plS38, plS39, and plS40, respectively.

**Plasmid plS41 and plS42**. The truncated gene *rs00400* fragments and truncated gene *rs02775* fragment from *S. suis* were amplified using the primers ol103/ol104 and ol105/ol106, respectively. The resulting PCR products were inserted into the corresponding site of pGEX-4T-1 digested by EcoRI and BamHI and constructed plasmids plS41 and plS42, respectively.

**Plasmid plS43**. The truncated gene *rs02775* fragment and gene *rs02780* from *S. suis* were amplified using ol107/0l108 and ol109/ol110, respectively. The resulting PCR products were combined by overlap extension PCR using the primers ol107/ol110. The resulting fragment was inserted into the corresponding site of pET-28a digested by EcoRI and BamHI, yielding the plasmid plS43.

**Plasmid plS44 and plS45**. The gene *rs02080* from *S. suis* was amplified using the primers ol111/ol112. The resulting fragments were inserted into the corresponding site of pGEX-4T-1 digested by EcoRI and BamHI, and constructed plasmid plS44. The gene *rs02075* from *S. suis* was amplified using the primers ol113/ol114. The resulting fragments were inserted into the corresponding site of pET-28a digested by EcoRI and BamHI, and constructed plasmid plS45.

**Plasmid plS46 and plS47**. The truncated gene *rs02080* fragment and truncated gene *rs00400* fragment from *S. suis* were amplified using ol115/ol116 and ol117/ol118, respectively. The resulting PCR products were inserted into the corresponding site of pET-28a digested by EcoRI and BamHI and constructed plasmid plS46 and plS47, respectively.

**Plasmid plS48-plS53**. The homology fragments of gene *rs00400* from *S. pyogenes*, *S. agalactiae*, *Streptococcus equine subspecies of zoonosis*, *S. suis*, *S. pneumoniae*, and *S. aureus* were amplified using ol119/ol120, ol121/ol122, ol123/ol124, ol125/ol126, ol127/ol128, and ol129/ol130, respectively. The resulting PCR products were inserted into the corresponding site of pGEX-4T-1 digested by EcoRI and BamHI, and constructed plasmids plS48-plS53, respectively.

**Plasmid plS54 and Plasmid plS55**. The whole plasmid sequence was amplified using the primers ol131/ol132 with plasmid plS51 as a template, and the resulting PCR product was digested by DpnI and constructed the plasmid plS54. Likewise, constructing the plasmid plS55 using the primers ol133/ol134.

**Plasmid plS56 and plS57**. To obtain the plasmid plS56. The upstream and downstream flanking regions of the gene *rs10215* from *S. agalactiae* were amplified using ol135/ol136 and ol137/ol138, respectively. The resulting two fragments were fused by a third PCR reaction using the primers ol135/ol138, and the resulting third fragment was inserted into the corresponding site of pSET4s digested with EcoRI and BamHI. Likewise, the upstream and downstream flanking regions of genes *rs06310* and *rs06305* were amplified from *S. agalactiae* using the primers ol139/ol140 and ol141/ol142, respectively, and construct corresponding plasmids plS57.

**Plasmid plS58**. The gene *rs10215* with its native promoter region from *S. agalactiae* was amplified using the primers ol143/ol144. The resulting fragment was inserted into the corresponding site of pSET2, digested with EcoRI and BamHI, and constructed the plasmid plS58.

**Plasmid plS59 and plS60**. The whole plasmid sequence was amplified using the primers ol145/ol146 and ol147/ol148 with plasmid plS58 as a template. The resulting PCR products were digested with DpnI, and plasmid plS59 and plS60 were constructed.

**Bacterial two-hybrid plasmids**. All primers for bacterial two-hybrid primers were named T25-N-F/R and T18-N-F/R. These primers were used to amplify the corresponding genes, and the related PCR products were inserted into pUT18C digested with EcoRI and BamHI or pKT25, digested with BamHI and KpnI.

## Protein production and purification

The general procedure for protein expression and purification: *E. coli* BL21(DE3) containing expression plasmid was grown in 1 L LB supplemented with ampicillin or kanamycin at 37 °C until the $OD_{600} = 0.4$–0.6. Then the protein expression was induced by adding 0.5 mM IPTG, and the culture was incubated at 16 °C overnight. Cells were harvested by centrifugation ($7000 \times g$, 10 min, 4 °C), and the pellet was stored at −80 °C. The pellet was resuspended in 25 mL buffer A (20 mM phosphate buffer, pH 7.4, 500 mM NaCl) supplemented with 1 mM PMSF, 20 µg mL$^{-1}$ lysozyme. After sonication and centrifugation ($10,000 \times g$, 10 min, 4 °C), the supernatant was loaded onto a HisTrap HP column (GE Healthcare) and washed with buffer A containing 20 mM imidazole. Proteins were eluted with buffer A containing 100–500 mM imidazole. The elution fractions were analyzed with 12% SDS gel. The pure protein fractions were pooled and dialyzed overnight in buffer B (20 mM Tris, pH 8.0, 150 mM NaCl) at 4 °C and concentrated using a 10 kDa molecular weight cut-off centrifugal filter device (Millipore). In addition, for the purification of GST-tag proteins, the supernatant, after sonication and centrifugation, was loaded onto a GSTrap HP column (GE Healthcare). Finally, buffer A, supplemented with 20 mM glutathione, was used to elute proteins.

For the purification of CcpS-P, *E. coli* BL21(DE3) was simultaneously transformed with plasmid plS30 and plS44 for co-expression of CcpS and Stk1. And colonies were incubated into LB supplemented with ampicillin and kanamycin. The expression and purification procedures were the same as above. The phosphorylation of pure CcpS was tested using 15% Phos-tag gel containing 50 µM Phos-tag solution (Fujifilm Wako pure chemical, AAL-107) and 100 µM MnCl$_2$ (Applicable to all percentages of acrylamide). Similarly, for the purification of CpsD-P, *E. coli* BL21(DE3) containing plasmid plS43 for expression of chimeric protein CpsC/D. And 12% Phos-tag gel analysis further confirmed the phosphorylation of CpsD. Protein small aliquots were stored at −80 °C.

For the crystallization of CcpS and its mutant, the expression and purification procedure were the same as above. Subsequently, the His-tag was cleaved by tobacco etch virus protease digestion in a mass ratio of 1:20 at 16 °C overnight in buffer B. The mixture was loaded onto the HisTrap HP column again, and the His-tag free CcpS or its mutant was recovered in the flow-through. The fractions' purity was analyzed by 15%

SDS gel and Coomassie staining. The protein samples were further purified by gel filtration on a Superose 10/300 GL column (GE) equilibrated in buffer B. The concentration of proteins was measured at 280 nm with a NanoDrop spectrophotometer (Thermo Fisher Scientific).

## Crystallization and structure determination

The CcpS and CcpS-T4ET7E variant were concentrated to 10 mg mL$^{-1}$ in buffer (20 mM Tris, pH 8.0, and 200 mM NaCl). The initial crystallization was screened using commercial kits, and the crystallization was performed at 277 K with the sitting-drop and vapor-diffusion method and 1 μl drops of protein solution and reservoir solution. Due to the low diffraction resolution of the initial screen crystal, the crystallization condition was optimized by grid screening of the pH, precipitant, and protein concentration. For CcpS, the diffraction quality crystals at 0.1 M Tris: HCl pH 8.0 15% (V/V) ethanol and 0.1 M HEPES: NaOH pH 7.8 2.0 M ammonium sulfate for CcpS-T4ET7E variant. Before data collection, crystals were soaked for 1–15 min in the mother liquor supplemented with 10% glycerol, then frozen in liquid nitrogen. X-ray diffraction data were collected on the beamline BL19U1 of Shanghai Synchrotron Radiation Facility (SSRF) at 100 K. Data integration and reduction were performed using XDS[70]. Initial phases were determined with PHASER[71] using the *E. faecalis* IreB structure as a search model PDB: 5US5 (https://www.rcsb.org/structure/5US5)[52]. Manual corrections were performed with the Coot[72], and refinement was carried out using REFMAC5[73] and Phenix[74]. PyMol 2.2 was used to prepare figures of the structures[75]. Data collection and refinement statistics are provided in Supplementary Table S4.

## Preparation of protein polyclonal antibodies

To generate mouse antibodies against *S. suis* proteins: Stk1, CcpS, CpsB, CpsD, MurA1, MurZ, and Groel, mice were immunized with each protein antigen supplemented with Freund's adjuvant (Sigma-Aldrich, F5881). The specificity of the obtained antibody was determined by western blot using the cell extracts of the WT strain and corresponding gene deletion strains as antigens. The antiserum was supplemented with 50% glycerol and stored at −20 °C.

## Bacterial two-hybrid assay

The bacterial two-hybrid assay follows the manufacturer's instruction (BACTH System Kit) and references[33,76] with the following modifications. Briefly, the whole genome of *S. suis* was extracted. After sonication and purification, resulting in random fragments of the genome in the size of 250 to 1500 bp. Then the DNA fragments were further modified by T4 DNA polymerase and T4 DNA polynucleotide kinase, and the resulting products were ligated into pUT18C, digested with SmaI, and dephosphorylated with CIAP. Finally, the mixtures were transformed into *E. coli* DH5α and the colonies were pooled as the genomic library of *S. suis*. To identify the potential partners of CcpS, about 0.3 μg of the library plasmid pool was added to 100 μl competent *E. coli* BTH101 cells harboring bait plasmids for the expression of either T25-CcpS or CcpS-T25. The cell suspension was appropriately diluted and plated on M63 agar plates and was incubated at 30 °C in the dark. Blue colonies were picked and identified by PCR, and the resulting PCR products were sequenced. Each plasmid pair was co-transformed into BTH101 component cells to test the interaction between proteins. And then, the positive colonies were picked and incubated in 1 mL LB supplemented with 50 μg mL$^{-1}$ kanamycin, 100 μg mL$^{-1}$ ampicillin, and 0.5 mM IPTG at 30 °C. Five microliters of each culture was spotted in triplicate onto LB agar plates supplemented with 50 μg ml$^{-1}$ kanamycin, 100 μg ml$^{-1}$ ampicillin, 0.5 mM IPTG, and 40 μg ml$^{-1}$ X-gal. Plates were incubated at 30 °C for 30 h, and the images were captured using a digital camera.

## Pull-down assay

To test the interaction between CcpS and CpsB. *S. suis* cells were grown in 20 mL THY medium at 37 °C until OD$_{600}$ = 0.5–0.6, and harvested by centrifugation at 5000×*g*, 10 min, 4 °C. Cells were frozen and thawed in liquid nitrogen. Then the cells were resuspended in 5 ml buffer C (buffer B supplemented with 0.1% Triton X-100), and 0.1 mM PMSF was added. After sonication, the cell lysate was centrifuged at 12,000×*g*, 10 min, 4 °C. The supernatant was pooled in ice. Subsequently, the supernatant was incubated with 100 nM rCpsB-GST (r means recombinant protein) at 4 °C for 4 h, and then incubated with 10 μl glutathione agarose beads for another 1 h at 4 °C on a rotator. An empty GST-tag was used as a control. After incubation, the beads were collected by centrifugation at 500×*g*, 5 min, 4 °C and washed five times with buffer C. Then the beads were resuspended in 20 μl SDS sample buffer and boiled for 10 min. Retained proteins were detected by Western blotting after 15% SDS gel.

To verify the direct interaction between CcpS and CpsB, 1 μM rCcpS-GST and 5 μl prewashed glutathione agarose beads were incubated in 600 μl buffer C at 4 °C for 0.5 h on a rotator, and then the beads were washed three times with buffer C. The beads were resuspended with 600 μl buffer C and incubated with 0.3 μM rCpsB-His at 4 °C for 10 min. Finally, the beads were washed five times with buffer C. The retained protein was detected by Western blotting after 12% SDS gel. A similar strategy was applied to determine the interaction between CcpS and CpsD.

## Kinase assays

The autophosphorylation of Stk1 and phosphorylation of CcpS were performed as described previously with the following modification[77]. About 1.5 μM rCcpS and 0.3 μM rStk1$_{1-346}$ (1–346 means truncated recombinant protein) were incubated in 100 μl phosphorylation buffer D (100 mM Tris, pH 8.0, 10 mM MgCl$_2$, 1 mM DTT, and 5 mM ATP) at 37 °C for 2 h. And rCcpS was individually incubated in buffer D as control. Then add 20 μl SDS sample buffer to the mixture and boiled for 10 min at 100 °C. The samples were loaded onto 12% Phos-tag gel and 12% SDS gel and Coomassie stain. The protocol for detecting CpsD kinase activity was adapted from previously published methods with modifications[51]. Briefly, 1.5 μM rCpsD was incubated in buffer D at 37 °C with the presence or absence of 3.5 μM rCcpS for 10 min in a total volume of 50 μl, then 1.5 μM rCpsC-ctr was added to each volume. The samples were incubated for 30 min at 37 °C, SDS sample buffer was added to quench the reaction, and samples were boiled at 100 °C for 10 min. The samples were loaded onto a 12% SDS gel and for Western blotting analysis.

## Phosphatase assays

This procedure was adapted from a previous report[78]. To measure the de-phosphorylation of CcpS, 1.5 μM rCcpS-P was incubated in buffer B supplemented with 1 mM MnCl$_2$ with the presence of 50 nM rStp1 or rCpsB at room temperature for 30 min in a total volume of 50 μl. Then 15 μl SDS sample buffer was added to quench the reaction, and samples were boiled at 100 °C for 10 min. The samples were loaded onto a 12% SDS gel and 12% Phos-tag gel, respectively, run at 140 V and Coomassie stain.

To analyze the de-phosphorylation of CpsD by CpsB, 1.5 μM rCpsB was preincubated in buffer B supplemented with 1 mM MnCl$_2$ with the presence of 3.5 μM rCcpS or rCcpS-P at room temperature for 10 min in a total volume of 50 μl. Then 1.5 μM rCpsD-P was added to each volume to initiate reactions. The samples were incubated for 30 min or removed at the indicated time at 37 °C. SDS sample buffer was added to quench the reaction, and samples were boiled at 100 °C for 10 min. The samples were loaded onto a 12% SDS gel and run at 140 V. CpsD phosphorylation was detected by Western blotting. All immunoblots represented at least three independent replicates.

Colorimetric phosphatase assays of CpsB activity were performed using p-nitrophenyl phosphate (pNPP), as described previously in ref. 78. Briefly, 50 nM rCpsB, rCpsB-R139A, or rCpsB-R206A were preincubated in buffer B supplemented with 1 mM MnCl$_2$ at 37 °C for 10 min. The reactions were initiated by adding 16 μM pNPP. The

relative activity of CpsB was measured according to the absorbance values at $OD_{405}$ for 10 min at 37 °C.

## CPS analysis

The cell surface sialic acid (SA) was released as previously reported with the following modification[47]. In brief, indicated cells were grown in 50 ml THY medium at 37 °C with shaking for 8 h, and the cells culture densities were normalized to an $OD_{600} = 1.0$. Subsequently, the same volume culture medium were collected by centrifugation at 5000×$g$, 4 °C for 10 min, and the cells' pellet was washed with PBS one time. Cells were resuspended in 0.8 ml PBS supplemented with 0.25 U neuraminidase (Solarbio, N7790), and the samples were incubated at 37 °C on a rotator for 1 h to release the cell surface SA thoroughly. Then the samples were collected by centrifugation at 10,000×$g$, 4 °C for 10 min, and the supernatant was filtered with 0.22-μm filters (Millipore). The content of SA in the digestive supernatant was determined by the colorimetric resorcinol-hydrochloric acid method[50].

Cross-absorbed anti-serotype 2 CPS antiserum was prepared as described previously in ref. 19. In brief, the cells of an unencapsulated strain Δ*cpsEF* (Δ*cps*) was grown in 500 ml THY medium at 37 °C with shaking for 8 h. Cells were collected by centrifugation at 5000×$g$, 4 °C, 10 min. The cells pellet was washed with PBS twice and inactivated by incubating it at 60 °C for 1.5 h. Then the heat-killed cells were resuspended in 10 ml PBS supplemented with 0.5 ml mouse anti-serotype 2 antisera. The mixture was incubated at 4 °C on a rotator overnight. The samples were collected by centrifugation at 10,000×$g$, 4 °C, 10 min, and the supernatant were filtered with a 0.22-μm filter (Millipore). The absorbed antiserum was stored at 4 °C or −20 °C, and supplemented with 50% glycerol. CPS detection as previously methods with the following modifications[50]. To detect the CPS releasing to medium, cells were grown in 8 mL THY at 37 °C for 8 h, and the indicated cells culture densities were normalized to an $OD_{600} = 1.0$ with the same volume adjusted by fresh THY, and 6 ml fractions were collected by centrifugation at 5000×$g$, 10 min, at room temperature. The supernatant was filtered with a 0.22-μm filter. Then the sample was treated by adding 5 μl proteinase K (Qiagen, 19133) and incubated at 56 °C for 1 h, and 10 μl serial dilution (1:2) samples were spotted onto a PVDF membrane and air dried. Subsequently, the membrane was fixed with 70% ethanol for 5 min and air dried again for 15–20 min. The membrane was blocked in 5% non-fat milk in PBS containing 0.05% Tween-20 for 0.5 h at room temperature, and cross-absorbed anti-serotype 2 CPS antiserum was added at a 1:150 dilution followed by incubation at room temperature for 2 h. The anti-mouse IgG conjugated to HRP (Engibody) as the subsequent secondary antibody at a 1:10,000 dilution, ECL color development, ChemiDoc Imaging system (Bio-rad), and all immunoblots represent at least three independent replicates.

## Phagocytosis assay

Mouse macrophage cells Raw264.7 were cultured in high-glucose Dulbecco's modified Eagle medium (DMEM) (Gibco) supplemented with 10% fetal bovine serum (Gibco) at 37 °C, 5% $CO_2$. The method of bacterial phagocytosis was described previously with the following modifications[56]. In brief, the WT and derivative strains were grown in THY at 37 °C until $OD_{600} = 0.4–0.6$. The cells were washed with PBS three times and were normalized to the same CFU resuspended in DMEM. Then 1 ml $10^7$ CFU bacteria suspension was added per well at a ratio of about 100 bacteria per macrophage. Phagocytosis was left to proceed for 1 h at 37 °C, 5% $CO_2$. Then the cell monolayers were washed with PBS three times and reincubated in fresh DMEM supplemented with 10 μg ml$^{-1}$ penicillin G and 100 μg ml$^{-1}$ gentamicin for 1 h to kill extracellular bacteria at 37 °C, 5% $CO_2$. It has been demonstrated that these antibiotics do not penetrate eukaryotic cells under these conditions, and this concentration of antibiotics was able to kill any remaining extracellular bacteria[79]. Finally, the cell monolayers were

washed with PBS 3 times again, and 1 ml of sterile distilled water was added to lyse the macrophages. The intracellular bacteria were determined by plating serial dilutions of the lysates on THY agar plates. The recovered bacterial CFU was counted, and the assay was performed on at least three biological replicates.

## Transmission electron microscopy

Strains were grown in THY medium at 37 °C until the $OD_{600} = 0.4–0.6$. Cells were harvested by centrifugation at room temperature, 2000×$g$, 10 min. The pellet was resuspended and fixed in 0.2 M phosphate buffer containing 2.5% glutaraldehyde (Sigma-Aldrich, G5882) overnight at 4 °C. Subsequently, the samples were washed with 0.2 M phosphate buffer two times and then fixed with 1% osmic acid fixed solution at room temperature for 1 h. The samples were dehydrated with propylene oxide for 10 min and embedded in epoxy resin overnight at 70 °C. The ultra-thin sections (50–70 nm) were prepared by an ultra-thin slicer. Sections were subsequently stained with 3% uranyl acetate. Finally, the fixed samples were imaged by a Hitachi h-7500 transmission electron microscope (Hitachi), and the thickness of the capsule was analyzed by Image J software.

## Immunofluorescence microscopy

*S. suis* was grown exponentially in THY broth, and 1 mL bacterial cells were collected by centrifugation at 2000×$g$, at room temperature for 10 min. Then, washed two times with sterile phosphate buffer saline (PBS). Subsequently, cells pellets were resuspended in 100 μL PBS and 5 μL transferred onto slides, air dried, fixed with 4% paraformaldehyde (Sigma-Aldrich) (pH 7.5) by incubating for 15 min at room temperature, rinsed three times with PBS, and allowed to air dry again. Then, 200 μL cross-absorbed mouse anti-serotype 2 CPS antiserum was added at a 1:100 dilution to cover all cell areas, and the mixture was incubated at room temperature for 90 min, washed three times with PBS, and CPS was detected by adding 200 μL goat anti-mouse IgG conjugated with Alexa Fluor 647 (Abcam, ab150115) at a 1: 200 dilution and incubated at room temperature for 30 min. For DAPI, 100 μL DAPI was added at 2 μg μl$^{-1}$ (Solarbio) and incubated for 10 min at room temperature. After a last wash with PBS, the slides air dried, and then 5 μL mounting medium was added and covered by a cover slip. Slides were visualized with a Zeiss AxioObserver Z1 microscope fitted with an Orca-R2 C10600 charge-coupled device (CCD) camera (Hamamatsu) with a 100×NA 1.46 objective. Images were collected with axiovision (Carl Zeiss). The images were analyzed using ImageJ (Fiji)[80] and the plugin MicrobeJ[81].

## Stressor sensitivity assay

The *S. suis* strains were treated with the following stressors. Antibiotics and autolysis assays were used to test whether the absence of the kinase Stk1 and CcpS affect *S. suis* cell wall synthesis. Autolysis of *S. suis* was determined using methods described previously in ref. 82 with some modifications. Briefly, indicated strains grown to an $OD_{600}$ 0.4–0.6 at 37 °C were centrifuged, washed in PBS, and resuspended to the original volume in PBS supplemented with 0.01% Triton-100. The initial $OD_{600}$ was measured (T0). The cultures were then incubated at 30 °C without shaking, and the $OD_{600}$ was recorded every 2 h for a period of 8 h. Autolysis for each strain was shown as a percentage of the initial $OD_{600}$ reading at T0. Minimal inhibitory concentrations (MICs) against cephalosporin were determined as described previously[83] with some modifications. In brief, cephalosporin with the highest concentration of 4.096 μg ml-1 and serial dilutions (1:2) in a 96-well cell culture plate with fresh THY medium. Indicated strains were cultivated in THY at 37 °C overnight, and normalized to the same $OD_{600}$. Subsequently, 5 μL cultures were added to 96-well cell culture plates and the plates were incubated at 37 °C for 1 day. The initial and after incubation $OD_{600}$ were measured T0 and T1, and then MICs were determined according to T1-T0. To explore the effect of stressors on the phosphorylation level of CcpS, the indicated strains were grown in 20 mL THY at 37 °C until

$OD_{600} = 0.4–0.6$. Subsequently, the cells were subjected to stressors treatments with PBS starvation, THY supplemented with negative serum (negative serum for *S. suis*) at a ratio of 1:1, THY supplemented with anti-serotype 2 antisera at a ratio of 1:1, pH 5.5 THY adjusted by hydrochloric acid, THY supplemented with 0.4 M NaCl, and THY supplemented with 50 mM $H_2O_2$. In brief, the cell pellets were resuspended in equal volume above solutions and incubated at 37 °C for 20 min, and then the cells were harvested by centrifugation at 5000×g, 4 °C for 10 min. The cell pellets were added with protease inhibitor and phosphatase inhibitor (MedChemExpress) and stored at −80 °C. To analyze the dynamic changes of CcpS phosphorylation, cells were taken at an indicated time when strains against starvation in PBS, and then cells pellet was collected by centrifugation at 5000 × g, 4 °C for 10 min and stored at −80 °C. For each experiment, at least three biological replicates were performed.

### Immunoprecipitation after special treatment
The WT and indicated mutant strains were grown in 20 ml THY medium at 37 °C with shaking until $OD_{600} = 0.4–0.6$. The bacterial pellet was harvested by centrifugation at 5000×g, 4 °C for 10 min. And then, the cells pellet or the cells from other special treatments were frozen and thawed in liquid nitrogen repeatedly three times. Then the samples were resuspended in 5 ml PBS supplemented with protease inhibitor and phosphatase inhibitor. The whole process was carried out on the ice. After sonication, the samples were collected by centrifugation at 10,000×g, 4 °C, for 10 min, and the supernatant was collected. Subsequently, the fractions were incubated with 10 μl indicated antibody coupled protein A/G agarose beads at 4 °C for 1 h on a rotator. The beads were washed with ice-cold PBS five times and then resuspended in 20 μl SDS sample buffer. The samples were boiled at 100 °C for 10 min and subjected to Western blot analysis.

### Detection of phosphorylation by Phos-tag gels
To reconstitute the Stk1-CcpS system in *E. coli*, two plasmids, pIS46 and pIS51 were co-transformed into *E. coli* BL21(DE3), the resulting strain, simultaneously expressed rCcpS-GST and rStk1$_{1-346}$-His. Moreover, two plasmids, pIS51 and pET-28a were co-transformed into *E. coli* BL21(DE3), the resulting strain, only expressed rCcpS-GST in the absence of Stk1. Indicated strains were grown in 10 ml LB at 37 °C with shaking until $OD_{600} = 0.4–0.6$, then, the protein expression was induced by adding 0.5 mM IPTG, and the culture was incubated at 16 °C for 4 h. The cells were harvested by centrifugation at 5000×g, 4 °C for 5 min, and the cells' pellet was resuspended in 5 ml PBS. After sonication, the samples were centrifugated at 10,000×g, 4 °C for 10 min, and the supernatant was incubated with 5 μl prewashed GST agarose beads (Solarbio, P2020). The beads were washed with buffer B five times, and the retained proteins were eluted with 50 μl buffer B supplemented with 20 mM glutathione. After centrifugation at 10,000×g, 4 °C for 5 min, the concentration of supernatant proteins was measured at 280 nm with a NanoDrop spectrophotometer and then the supernatant was added with 15 μl SDS sample buffer, and boiled at 100 °C for 10 min. The samples were loaded onto 12% Phos-tag gel and SDS gel and Coomassie stain. For the immunoprecipitated samples from whole-cell lysates of *S. suis*, the 10 μl samples were loaded onto 15% Phos-tag gel. The gel was soaked in transfer buffer containing 5 mM EDTA and gently shaken for 10 min three times. Then the gel was soaked in transfer buffer without EDTA for 10 min. Subsequently, the samples were transferred to a PVDF membrane for Western blotting analysis. For each experiment, a minimum of three biological replicates were performed.

### Western blotting
The cells pellet, or the cells from other special treatments were frozen and thawed in liquid nitrogen repeatedly three times. Then the samples were resuspended in 5 ml PBS supplemented with protease inhibitor and phosphatase inhibitor (MedChemExpress, HY-K0021). The whole process was strictly carried out on the ice. After sonication, the

samples were collected by centrifugation at 10,000×g, 4 °C, for 10 min. Lysate corresponding to 25 μg protein or samples from other assays were boiled for 10 min at 100 °C in SDS sample buffer and resolved on 12% SDS gel or 15% Tris-Tricine SDS gel for small proteins and run at 140 V. Proteins were transferred onto polyvinylidene fluoride membrane (PVDF, Millipore) using a Semi-Dry Transfer Device (Bio-Rad) and run at 10 V for 20 min. The membrane was blocked in 5% non-fat milk or 1% bovine serum albumin for phosphorylation detection in PBS containing 0.05% Tween-20 for 0.5 h at room temperature. Proteins were detected using the following antibodies: Mouse anti-phosphotyrosine antibodies (Abcam, ab10321) at a 1:1000 dilution, Rabbit anti-phosphothreonine antibody (Cell Signaling Technology, 9381 S) at a 1:1000 dilution, Mouse anti-His-tag antibodies (Engibody, AT0025) at a 1:5000 dilution, Mouse anti-GST-tag antibodies (Engibody, AT0098) at a 1:5000 dilution, and mouse polyclonal anti-serum against Stk1, CcpS, CpsB, CpsD, MurA1, MurZ, and Groel at a 1:1000 dilution. Anti-rabbit IgG or anti-mouse IgG conjugated to HRP (Engibody) were used as secondary antibodies at a 1:10,000 dilution, goat anti-mouse IgG conjugated with Alexa Fluor 647 at a 1:200 dilution. ECL color development, ChemiDoc Imaging system (Bio-rad), and immunoblots represent at least three independent replicates.

### Surface plasmon resonance analysis
Surface plasmon resonance experiments were performed at 25 °C using a Biacore T200 system (GE Life Sciences, USA). rCpsB, rCpsD-3YE, and rCpsD-3YF were immobilized on the CM5 sensory chip at 600 response units (RUs). An uncoated "blank" channel was used as a negative control. All materials were exchanged to or dissolved in Hepes buffer [10 mM Hepes (pH 7.4), 150 mM NaCl, 3 mM EDTA, and 0.05% P20]. Ligands were diluted in running buffer and injected at different concentrations at a flow rate of 30 μl min$^{-1}$ at 25 °C. The kinetic parameter analyses were performed using Biacore T200 Evaluation Software with a 1:1 Langmuir binding model.

### Molecular docking
Homology modeling, as previously reported in ref. [84], *S. suis* CpsB structure was built from Cps4B from *S. pneumoniae* PDB code: 2WJF (https://www.rcsb.org/structure/2WJF) as a template using Discovery Studio 3.5. Align Sequences module was utilized to align two sequences. All parameters were set as default. CpsB structure was created using the Build Homology Models module in Discovery Studio 3.5. The optimization level was set as high. Total Energy and the lowest DOPE score was subjected to further energy minimization. The initial 11 residues of CcpS and its mutant were used as ligands since the total protein was too large to conduct a docking study. To investigate the interacting patterns between ligands and CpsB, we performed molecular docking to screen the orientation pattern of ligands in the binding putative-pocket of CpsB and identify the residues involved in the interaction. Autodock Vina[85] was used for semi-flexible docking, and all ligands adopted the same parameters for docking. Outputting the top nine conformations for each ligand. The most reliable binding poses were selected according to the interaction energy. All results were analyzed and visualized using PyMOL.

### Mouse virulence assay
The WT and ΔccpS mutant strains were grown in THY medium at 37 °C with shaking until $OD_{600} = 0.4–0.6$. The bacterial pellet was harvested by centrifugation at 5000×g, 25 °C for 10 min, the cells pellet washed twice and resuspended with PBS. And then, 6–8 weeks old female BALB/c mice were inoculated intraperitoneally with ~$10^8$ colony-forming units of WT strain and ΔccpS mutant strain. Mouse survival was followed for 5 days. Virulence assays were conducted three times with similar outcomes and the data correspond to groups of six mice. All mice were housed in a specific pathogen-free facility. Dark and light were cycled every 12 h. The ambient temperature was 20–26 °C, and the humidity was 40–60%. All animal experiments were approved by the Laboratory Animal Welfare

and Ethics Committee of Nanjing Agricultural University, China (approval number NJAU.No20210510065 and NJAU.No20211005144). The Chinese National Laboratory Animal Guideline for Ethical Review of Animal Welfare adhered to animal care and protocol.

## Protein sequence alignment and analysis

The analysis of the Stk1 domain was predicted by Interpro (ebi.ac.uk/interpr), and its transmembrane was analyzed by Phobius (phobius.sbc.su.se) online. Model schematics generated by Microsoft PowerPoint (ScienceSlides). The CpsB and CpsD protein sequence were aligned against all homology sequences by the blastp (blast.ncbi.nlm.nih.gov), and the selected corresponding homology sequences from *S. suis*, *S. aureus*, *S. pneumoniae*, and *S. agalactiae* were aligned using Clustal Omega (ebi.ac.uk/Tools/msa/clustalo) and presentation of alignment was performed using software Jalview with conserved or active sites were marked[86,87]. The CcpS protein sequence was aligned against all homology sequences for different Firmicutes families by blastp (blast.ncbi.nlm.nih.gov). The sequences with the highest identity were used to construct an initial phylogenetic tree by software mega 7.0, and the phylogenetic tree is presented by itol (itol.embl.de). Conservation visualization was obtained using WebLogo 3 (weblogo.threeplusone.com). The homology sequences of CcpS from *S. suis, Staypholococcus aureus, S. pneumoniae, S. agalactiae, S. pyogenes*, and *E. faecalis* were aligned using CLUSTALW (genome.jp/tools-bin/clustalw), PSIPRED (bioinf.cs.ucl.ac.uk/psipred) for secondary structure analysis and presentation of alignment was performed using ESPript 3[88].

## Statistics and reproducibility

For Western blot and Dot blot analysis, images are one representative of at least three biological repeats. Antibiotic susceptibility testing data (MICs) from at least three independent biological repeats with similar results. Bacterial two-hybrid assay with three independent biological repeats were performed. For Figs. 1e, f, 4a, b, d, 7a and Supplementary Figs. 1d, f, g, 4c, d, h, i, 7c, the experiments were repeated independently three times with similar results. Comparison between groups was done using one-way or two-way ANOVA followed by Bonferroni or Tukey's post-tests as indicated. Other experiments were analysed using unpaired $t$-tests. ND, non-detected. Statistical analyses were performed using GraphPad Prism version 8.2.1.441 software and the number of independent biological experiments are presented in the figure legends. Data were presented as mean ± SD. $P$ values less than 0.05 were considered statistically significant.

## Reporting summary

Further information on research design is available in the Nature Portfolio Reporting Summary linked to this article.

## Data availability

*S. suis* ZY05719 genome can be found at National Center for Biotechnology Information with the accession number GI: 820722437 (https://www.ncbi.nlm.nih.gov/nuccore/820722437). The structure-related data of CcpS and its mutant *S. suis* generated in this study have been deposited in the RCSB Protein Data Bank under accession code: 7Y8Z (https://www.rcsb.org/structure/unreleased/7Y8Z) and 7Y86 (https://www.rcsb.org/structure/unreleased/7Y86), respectively. The data supporting the findings of this study are available in this article and its Supplementary files, or from the corresponding authors upon request. Source data are provided with this paper.

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

## Acknowledgements

We thank the staff at the beamline BL19U1 of SSRF and the National Center for Protein Science Shanghai for assistance with diffraction data collection. The present study is supported by the National Key Research and Development Program of China (grant no. 2021YFD1800404 to H.F.).

## Author contributions

J.T. carried out the majority of the experimental work and the data analyses. M.G., M.C., and B.X. contributed to the experimental work. W.W., T.R., and J.T. determined the structures and related analysis. H.F. designed the research and experiments. J.T. carry out the bioinformatics analysis. Z.M. and H.L. contributed to the experimental design. J.T. wrote the manuscript and H.F. contributed to the manuscript. All authors read and commented on the manuscript.

## Competing interests

The authors declare no competing interests.
