## [Peer Review File · Nature Communications]

A link between STK signalling and capsular polysaccharide synthesis in *Streptococcus suis*Reviewer #1 (Remarks to the Author):

The present study reports the identification and the characterization of CppS in *S. suis*. It is shown that this protein controls CPS production. It is also demonstrated that CppS is phosphorylated by the S/T-kinase Stk-1 and that its dephosphorylation modulates the activity of the Y-phosphatase CpsB that is specifically involved in CPS synthesis. Together with other experiments, showing that CppS phosphorylation/dephosphorylation affects macrophage phagocytosis and impact both stress tolerance and CpsD dephosphorylation by CpsB, it is concluded that CppS is part of a signal transduction system interconnecting the CpsD and Stk1 regulatory networks.

The experiments are performed carefully and are interesting for a broad readership. However, some statements require much more work to be fully conclusive and to bring much more novelty. I think that some important controls and complementary experiments should be performed. In addition, a number of quantifications and statistical tests are also needed.

In this context, I encourage the authors to take my comments into account before resubmitting a revised version of the manuscript.

Major comments:

- Lines 80-81: I'm actually confused with this statement. It is not clear if the comparative analysis comes from this work or from a previous study? Please fix. If this analysis is part of this work, the phosphoproteomic data should be shown as a supplemental table providing the full list of proteins phosphorylated in WT and Δ stk-1 strains and highlighting phosphorylation events specific of Stk1. It could also be interesting to discuss some of them with regards to the literature and this present study. Statistic tables and spectra should also be provided. These data should also be deposited in a databank (ProteomeXchange Consortium for instance). Was the whole sequence of CppS covered? i.e. can the author exclude additional *in vivo* phosphorylation sites for CppS?

-Line 81 and related to other results: Why the author named this protein CppS? My understanding is that CppS is the homolog of IreB from *E. faecalis* (Hall et al, 2013, Antimicrob Agents Chemother and 2017, J Mol Biol) and ReoM from *L. monocytogenes* (Wamp et al, 2020, Elife, Kelliher et al, 2021, PLoS Pathog, Wamp et al, 2022, PLoS Pathog). These articles should be cited and the present results should be discussed in light of these articles. The authors should also consider doing some experiments to check if similar conclusions could be made. Notably, ReoM phosphorylation controls peptidoglycan synthesis through control of the ClpCP-dependent proteolytic pathway. Notably, the stability of MurA (that is also found to interact with CppS (MurZ, Supp Fig 3A) is controlled by the dephosphorylation of ReoM). In this context, an accurate analysis of the impact of CppS deletion on cell morphogenesis should be provided. Moreover, does cppS deletion affect resistance to cephalosporin? How the authors explain that cppS deletion does not impact antibiotic sensitivity (Supp Fig 2e). How the authors explain that deletion of both cppS and stk1 increases phagocytosis (Supp Fig 2f) while double deletion of reoM and prkA (homolog of stk-1 in *L. monocytogenes*) restores virulence? does CppS deletion reduce virulence?

- Interactions assays (B2H and pull-down): To be fully convincing, especially to conclude that the different mutations (phospho-ablative (by the way why a T to V substitution instead of a classical T to A were performed?), phospho-mimetic and dimerization) affect/modulate the interactions, appropriate biochemical approaches (microscale thermophoresis, SPR,...) should be performed and affinity constants should be provided).

- Lines 141-145: these data shows that phosphorylation of T7 is the physiologically relevant event. How the authors explain that? Is there some cooperativity between the phosphorylation of T7 and T4?

- lines 162-163: These experiments are not fully convincing. Additional images should be shown. A statistical analysis made with many cells is also necessary. I know that such statistics are difficult with TEM images as it requires many images but maybe the authors should consider using another method (quantification of the fluorescence signal of living cells immuno-detected with an anti-CPS antibody). This approach would also help in demonstrating if cell morphogenesis is affected (based on the analysis of phase contrast images). Related to these points, is the cell shape of the CpsD-

3YF mutant affected (lines 251-253). This mutation induces cell elongation and delocalization of the CPS production. It is important to conclude that phenotypes caused by CpsD and CppS are similar (line 252).

- Line 257: Is CpsD-CppS interaction modulated by CpsD phosphorylation?

- There are some typos and English errors that should be corrected through the manuscript. For instance, line 142, 215, 216, 399-400, ...)

Minor comments:

- Abstract, lines 18-19: it is already established that CpsB and CpsD are indispensable in Streptococci (and in firmicutes actually). I would change for "we also confirm that...."
- Introduction, lines 31-32: the "majority" is actually not fully accurate. I would be better to simply state that there are 2 main systems, the the ABC-transporter dependent system and wzy system, the latter being the most widespread. Please cite Whitfield et al, 2020, Annu Rev Microbiol.
- A series of appropriate references (more recent and/or more related to Streptococci and/or alternative conclusions) are missing: Line 41: please cite Jadeau et al, 2008, Bioinformatics as reference for the BY-kinase family. Line 53: please cite as well Ducret&Grangeasse, 2021, Curr Opin Microbiol. Lines 59-60: Please cite Nagarajan et al, 2022, Trends Microbiol. Line 63: Please cite Fleurie et al, 2014, PLoS Genet. Line 225: Please cite Nourikyan et al, 2015, PLoS Genet.
- Supp Fig 2d: Not easy to read. A color code could be helpful.

Reviewer #2 (Remarks to the Author):

The manuscript by Tang and co-workers describes a substantial body of work to understand the regulation of capsule synthesis in *Streptococcus suis*. They report a new link between the Stk1 kinase and capsule production mediated by interaction between the CcpS kinase substrate and the CpsB capsule synthesis regulator. This new connection is of interest and expands the landscape of physiological outputs that are known to be regulated by Stk1 homologs in bacteria.

The most novel and impactful findings reported in this manuscript are the connection between CcpS and CpsB and the resulting influence on CpsBCD phosphorylation and capsule synthesis. Many of the other findings reported here that relate to: (1) Stk1 phosphorylation of CcpS at T4 and T7; (2) impact of mutations at T4 and T7 on CcpS function; (3) the structure of CcpS and the existence of the flexible N-terminal tail; (4) impact of mutations at the dimer interface on CcpS; and (5) alterations of CcpS phosphorylation in response to environmental stimuli (suspension in PBS in particular), all have been reported previously for homologs of Stk1/CcpS in other bacterial species (*Enterococcus* mainly, but also *Listeria* to some extent) and therefore serve mainly to confirm in *S. suis* what has already been observed. Of course it is important in this manuscript to demonstrate similar results experimentally in *S. suis* for these results to be rigorous overall, but the authors never cite the previous work and instead present their findings as though nothing was previously known about this kinase/substrate signaling pair (or the CcpS 'family' of proteins), when that is in fact not the case.

Line 80-81: The authors refer to a "comparative analysis of phosphoproteins" between wild-type and Stk1-depleted cells that led them to investigate CcpS. However, other than CcpS, no data from this analysis is presented. Was CcpS the only differentially phosphorylated protein identified? If not, it would be of interest to see a list of all the differentially phosphorylated proteins that could be added to the supplemental material.

Fig 1f (and several other IP figures also): Immunoblotting was performed on IP samples to analyze CcpS phosphorylation, and GroEL immunoblots were included as "loading control". It is unclear how GroEL could be used as a "loading control" for such samples – was GroEL also IP'd with the anti-CcpS antibody?

Line 232: There appears to be a typo; it appears the mutations were made in *cpsD*, not *ccpS*.

Lines 334-335: The authors state that "CcpS phosphorylation level decreased rapidly and maintained a low level over time" in regards to figure 4f, which seems misleading because it appears from the figure that overall phosphorylation level actually remains relatively constant while the amount of total CcpS protein increases upon PBS challenge. In other words, the existing CcpS may not have necessarily been dephosphorylated after PBS challenge, but rather that new CcpS was synthesized and did not become phosphorylated. Or at the very least, it is not possible to distinguish between these possibilities on the basis of the data presented.

Fig 4: The legend title indicates that Stk1/CcpS "improves the stress tolerance of *S. suis*", but this is not justified because there is no data to demonstrate this.

Lines 476-477: The authors state that Stk1/CcpS control of capsule synthesis "enhances the survival ability of bacteria against the stress of the host immune system". This is an overstatement of the data presented that is not justified. While the data do show an effect of Stk1/CcpS on survival in macrophages, whether or not that survival effect is specifically the result of an influence on capsule synthesis has not yet been established experimentally through analysis of appropriate double mutants.

Line 485: The authors state that "Stk1 is conserved almost in the whole field of bacteria", which is inaccurate. Homologs of Stk1 – in other words, kinases with extracellular PASTA domains - are almost ubiquitous among Firmicutes and Actinobacteria, but they are rare among Gram-negative bacteria.

Line 625: It appears there should be a reference here to *S. aureus*, and not *E. faecalis*, based on the data presented.

Experimental conditions for Phos-tag gels (including acrylamide percentage, phos-tag concentration, divalent cation identity and concentration) should be included for all.

Lines 795-807 (phagocytosis assay): Have controls been performed to establish that the antibiotic regimen used is sufficient to kill extracellular *S. suis* bacteria? No such controls are described or shown. If this is known from previous work, a citation should be included.

Supplementary figure 1c: Appropriate controls are not included. We cannot rule out the possibility that the band labeled as CcpS-P is actually Stk1.

Supplementary figure 1f: I was unable to discern which line corresponded to which sample due to the low resolution (low resolution was an issue for many of the figures). Also, the legend states 3 replicates were performed: are the averages plotted? I do not see any mention of statistical analysis.

Supplemental figure 3a: I performed searches at NCBI for several of the gene numbers listed. Although I did not test all genes listed, the results I obtained did not correspond with the annotations given in the table. Please correct.

Reviewer #3 (Remarks to the Author):

Tang et al has illustrated the role of conserved Coordinator of Capsular Polysaccharide Synthesis (CcpS) playing a crucial role in Capsular polysaccharide (CPS) synthesis, which itself is pivotal to virulence. Using biochemical methods, the authors have shown that CcpS is a substrate of serine/threonine kinase Stk1. They have shown CcpS play the role of a regulatory hub in connecting bacterial tyrosine kinase system CpsBCD into the STK.

1. This system is very similar to the IreK-IreB protein regulatory network in gram positive bacteria (reference below). Could the authors highlight any unique differences between them ?
Hall, C. L., Tschannen, M., Worthey, E. A., & Kristich, C. J. (2013). IreB, a Ser/Thr kinase substrate, influences antimicrobial resistance in *Enterococcus faecalis*. *Antimicrobial agents and chemotherapy*, 57(12), 6179-6186.
2. The above paper also mentions the role of Threonine residues as substrates for STK. Do the authors using molecular modeling or simulation methods show the structural basis for the interaction and catalysis.
3. Although the crystal structure is solved for the WT and mutant CcpS, no structural analysis is performed and figure 5F, is not very clear in what authors want to represent. The justification of structure is not clear nor it gains in new insights into the substrate interaction or the role of mutation. Either a MD simulation or a complex of CcpS with STK1 could shed some more insights.
4. Overall, the quality of figures are subpar and it would be great to improve the figures in the further corrections.

Response to the Reviewers' comments:

Dear Reviewers,

The authors would like to thank for the Reviewers' constructive comments concerning our manuscript entitled "Phosphorylation-dependent CcpS controls CPS synthesis in Streptococci by modulating CpsB activity" (ID: NCOMMS-22-21790). These comments are all valuable and help us improve our manuscript. We have added more experimental data and modified the manuscript as recommended. We have made major adjustments and the manuscript was substantially revised. **All main changes were highlighted in red in the revised manuscript.** In addition, the point-to-point responses to the Reviewer's comments are as following:

In summary (main revisions):

1. We have done some experiments, including cephalosporin resistance (Supplementary Fig. 3b), bacterial autolysis (Supplementary Fig. 3c), and MurA1 and MurZ protein levels (Supplementary Fig. 3d-g) to check whether CcpS has similar functions with IreB/ReoM, as reviewer 1 suggested.
2. We performed surface plasmon resonance experiments to analyze the CcpS variants affinity to CpsB (Fig. 5e, f), we also tested the rCpsD-3YE and rCpsD-3YF interactions with rCcpS (Supplementary Fig. 6a-d), as reviewer 1 suggested.
3. As reviewer 1 suggested, we have compared the cell-associated CPS fluorescence signal (Fig. 6g, h) and cell morphogenesis. Statistical analysis have made with enough cells.
4. We have done some experiments and cited the previous work, and also further discuss related findings at proper position in the revised manuscript, as reviewer 2 suggested.
5. We also performed Western blot assay to detect total CcpS protein level at the same time, And we have corrected it in this figure (Supplementary Fig. 2a, b), as reviewer 2 suggested.
6. As reviewer 3 suggested, we have added the structural analysis at proper position in the revised manuscript (Supplementary figure 1h) (lines 482-495). In addition, we have performed molecular docking for CpsB and CcpS (figure 5c, d).
7. We have revised the statement as "We identified a potential Stk1 specific substrate hypothetical protein RS00400 (renamed CcpS) by phosphoproteomic analysis in a previous study. We will clarify this point in the revised manuscript (lines 102-104)", as reviewer 1 and 2 suggested.
8. We have entitled manuscript as "Evidence for a link between STK signalling and Wzx-Wzy pathway for CPS synthesis in bacteria", and we have rewritten abstract section.
9. We have revised the introduction as reviewers suggested and rewritten conclusions section.
10. **Panel order** has been rearranged in all result sections and also supplementary files panel, so that manuscript are easier for readers to follow. We have checked all the figures and ensured high resolution in the revised manuscript, as reviewers suggested.
11. **Please note** that there is an excess of data not necessarily germane to the story at hand, in order to make it easier and clearer for readers to follow, we have removed some redundant data, including Fig 2d, Fig 2i, Fig 2j, Fig 2k, Fig 3e, Fig 3i, Fig 3g, Fig 3h, Fig 4h, Fig 5d, and Fig 5f **in original manuscript** and also Supplementary Fig. 2b, Supplementary Fig. 2c,

Supplementary Fig. 2h, i, j, Supplementary Fig. 4d, e, f, and Supplementary Fig. 5h, k, l, m, n.

In addition, we have added some data, including Fig 1h, Fig 2b, d, f, h, Fig 3c, e, g, Fig 4g, i, Fig 5, Fig 6b, d, e, f, g, h, and Fig 8 **in the revised manuscript** and also Supplementary Fig. 1d, Supplementary Fig. 2b, e, Supplementary Fig. 3d, e, f, g, k, Supplementary Fig. 4j, k, m, Supplementary Fig. 5b, d, e, g, Supplementary Fig. 6, Supplementary Fig. 7a, b, f, g, Supplementary Fig. 8d, f, h, and Supplementary Fig. 9e, f. Of course, all figure legends have been changed accordingly.

12. We have substantially restructured manuscript, **please note the following major changes:** We added two sections (Fig 5 and Fig 6) in results. In other words, the original manuscript has 5 result sections, while there are 7 result sections in the revised manuscript. In addition, the result section 4 in original manuscript has been adjusted to second result section in the revised manuscript and the supplementary figures have also been changed accordingly. In addition, construction of plasmids has been adjusted and summarized in Supplementary Methods (lines 616-718). Also, the description of all the result sections have been changed accordingly.

13. We have rewritten the Discussion section, and “A model schematic depicting the mechanism of CPS synthesis coordination by CcpS phosphorylation” has been separated as Fig 8.

14. We added Tingting Ran in the author list and all authors have approved.

15. We addressed all other issues raised by the three reviewers and made corrections for trivial mistakes. **All main changes were highlighted in red in the revised manuscript.**

Reviewer #1 (Remarks to the Author):

The present study reports the identification and the characterization of CppS in *S. suis*. It is shown that this protein controls CPS production. It is also demonstrated that CppS is phosphorylated by the S/T-kinase Stk-1 and that its dephosphorylation modulates the activity of the Y-phosphatase CpsB that is specifically involved in CPS synthesis. Together with other experiments, showing that CppS phosphorylation/dephosphorylation affects macrophage phagocytosis and impact both stress tolerance and CpsD dephosphorylation by CpsB, it is concluded that CppS is part of a signal transduction system interconnecting the CpsD and Stk1 regulatory networks.

The experiments are performed carefully and are interesting for a broad readership. However, some statements require much more work to be fully conclusive and to bring much more novelty. I think that some important controls and complementary experiments should be performed. In addition, a number of quantifications and statistical tests are also needed.

In this context, I encourage the authors to take my comments into account before resubmitting a revised version of the manuscript.

Response: We sincerely thank the Reviewer's positive evaluation for our manuscript. The constructive comments and suggestions have helped us improve the quality of our manuscript. We added more experimental data and corrections **in the revised manuscript.**

Major comments:

- Lines 80-81: I'm actually confused with this statement. It is not clear if the comparative analysis comes from this work or from a previous study? Please fix. If this analysis is part of this work, the phosphoproteomic data should be shown as a supplemental table providing the full list of proteins phosphorylated in WT and Δ stk-1 strains and highlighting phosphorylation events specific of Stk1. It could also be interesting to discuss some of them with regards to the literature and this present study. Statistic tables and spectra should also be provided. These data should also be deposited in a databank (ProteomeXchange Consortium for instance). Was the whole sequence of CppS covered? i.e. can the author exclude additional *in vivo* phosphorylation sites for CppS?

We apologize for the vagueness of this statement. We identified a potential Stk1 specific substrate hypothetical protein RS00400 (renamed CcpS) by phosphoproteomic analysis in a previous study. We will clarify this point in the revised manuscript (lines 102-104).

And we detected a phosphorylated version of CcpS-specific peptides, but not the whole sequence of CcpS. In addition, we have excluded any additional *in vivo* phosphorylation sites for CcpS by a series of experiments (Fig. 1f, Supplementary Fig. 1f, g). We have observed that Stk1 and CcpS form an interaction complex and the kinase Stk1 specifically phosphorylates CcpS at residues Thr4 and Thr7 (lines 111-142).

-Line 81 and related to other results: Why the author named this protein CcpS? My understanding is that CppS is the homolog of IreB from *E. faecalis* (Hall et al, 2013, Antimicrob Agents Chemother and 2017, J Mol Biol) and ReoM from *L. monocytogenes* (Wamp et al, 2020, Elife, Kelliher et al, 2021, PLoS Pathog, Wamp et al, 2022, PLoS Pathog). These articles should be cited and the present results should be discussed in light of these articles. The authors should also consider doing some experiments to check if similar conclusions could be made. Notably, ReoM phosphorylation controls peptidoglycan synthesis through control of the ClpCP-dependent proteolytic pathway. Notably, the stability of MurA (that is also found to interact with CppS (MurZ, Supp Fig 3A) is controlled by the dephosphorylation of ReoM). In this context, an accurate analysis of the impact of CppS deletion on cell morphogenesis should be provided. Moreover, does cppS deletion affect resistance to cephalosporin? How

the authors explain that cppS deletion does not impact antibiotic sensitivity (Supp Fig 2e). How the authors explain that deletion of both cppS and stk1 increases phagocytosis (Supp Fig 2f) while double deletion of reoM and prkA (homolog of stk-1 in *L. monocytogenes*) restores virulence? does CcpS deletion reduce virulence?

We thank the referee for pointing this out, CcpS is really the homolog of IreB from *E. faecalis* and ReoM from *L. monocytogenes*. However, our results suggested that the Stk1-CcpS system is required for CPS production in *S. suis* (Fig. 3). Importantly, CcpS activity was tuned by phosphorylation mediated by Stk1/Stp1 system. Our data showed that the CcpS function is different from IreB/ReoM, whose functions mediated by regulated proteolysis. Thus, we renamed RS00400 as CcpS (Coordinator of capsular polysaccharide Synthesis) (lines 240-314).

We agree with the reviewer's comments on these articles that we have not fully discussed

them in our manuscript. And we will add and fully discuss these references at proper position in the revised manuscript (lines 246-270).

We have done some experiments, including cephalosporin resistance (Supplementary Fig. 3b), bacterial autolysis (Supplementary Fig. 3c), and MurA1 and MurZ protein levels (Supplementary Fig. 3d-g) to check whether CcpS has similar functions with IreB/ReoM. And all analysis and discussion have been included in the revised manuscript (lines 245-270).

In addition, we also observed that no significant impact of CcpS deletion on cell morphogenesis. This data is shown below, but not included in the revised manuscript (**Here**, we are very sorry for that some data were not all included in the revised manuscript temporarily, because an excess of data not necessarily germane to the story at hand. And we tried to make it clearer and easier for readers to follow.).

Our data showed that CcpS and its phosphorylation have no effect on cephalosporin resistance in *S. suis* (Supplementary Fig. 3b). This is a very interesting question that CcpS homologs IreB/ReoM are related to bacteria resistance to cephalosporin in *E. faecalis* and *Listeria monocytogenes*^{1,2}, while CcpS and its phosphorylation are negligible for *S. suis* resistance to cephalosporin.

To explain it, we want to know whether CcpS functions also mediated by regulated proteolysis thereby regulating cephalosporin resistance². Notably, both MurA1 and MurZ proteins levels were present at WT strain levels in *ccpS-T4ET7E* and *ccpS-T4VT7V* mutants strains (Supplementary Fig. 3d-g), which suggested that the function of CcpS phosphorylation may not mediated by regulated proteolysis (lines 245-270).

For further proof, we do some additional experiments. This data is shown below, but not included in the revised manuscript. We observed that single $\Delta murA1$ or $\Delta murZ$ mutant strain has no significant effect on *S. suis* resistance to cephalosporin. Importantly, we found that a single $\Delta murA1$ or $\Delta murZ$ mutant strain was easily obtained, while double *murA1- murZ* deletion was lethal to *S. suis*, which indicated that MurZ and MurA1 may be functionally replaced in *S. suis*. Previous studies also showed that MurZ and MurA homologs can be functionally replaced in *S. aureus* and *S. pneumoniae*^{3,4}, so that proteolytic degradation of the primary enzyme can be tolerated². Thus, we concluded that CcpS phosphorylation proposed not mediated by regulated proteolysis, and thus CcpS deletion and phosphorylation did not impact antibiotic sensitivity in *S.*

suis (lines 790-805).

Drug name	MIC($\mu\text{g}/\text{mL}$)			
	WT	ΔccpS	ΔmurZ	ΔmurA1
Ceftazidime (BS)	2.048	2.048	1.024	2.048
Ceftriaxone (BS)	0.256	0.256	0.256	0.256
Cefuroxime (ES)	0.256	0.256	0.256	0.256

BS, broad spectrum; ES, expanded spectrum; NS, narrow spectrum

Our data indicated that the Stk1-CcpS system is required for bacteria to resist phagocytosis and CPS production in *S. suis*. We only tested the phagocytosis rate of $\Delta\text{stk1}\Delta\text{ccpS}$, but not cytosolic survival, and we speculate that the increased phagocytosis rate of $\Delta\text{stk1}\Delta\text{ccpS}$ may be related to decreased CPS production (lines 271-314) according to our research. While the previous study showed that ReoM facilitates cytosolic survival in *L. monocytogenes*⁵. In addition, our previous data showed that CcpS deletion reduce virulence in animal models. Also, we speculate that the reduced virulence may be associated with decreased CPS production of ΔccpS mutant strain according to our research. This data is shown below, but not included in the revised manuscript.

- Interactions assays (B2H and pull-down): To be fully convincing, especially to conclude that the different mutations (phospho-ablative (by the way why a T to V substitution instead of a classical T to A were performed?), phospho-mimetic and dimerization) affect/modulate the interactions, appropriate biochemical approaches (microscale thermophoresis, SPR,...) should be performed and affinity constants should be provided).

Thanks for the helpful comments. To address this, we performed surface plasmon resonance experiments, rCpsB was immobilized on the CM5 sensory chip and relative SPR response unit (RU) was induced by rCps variants in a dose-dependent manner (Fig. 5e, f). Furthermore, we also tested the rCpsD-3YE and rCpsD-3YF interactions with rCpsS (Supplementary Fig. 6a-d). All results have been incorporated in the revised manuscript at proper position (lines 520-529). In addition, we used the T to V but not T to A substitution because we think that valine bear greater structural similarity to threonine than does alanine.

- Lines 141-145: these data shows that phosphorylation of T7 is the physiologically relevant event. How the authors explain that? Is there some cooperativity between the phosphorylation of T7 and T4?

Thanks for your comments. Our data showed that the single T4V mutations has no effect on CPS production in original manuscript. This data is shown below, but not included in the revised manuscript. Furthermore, our data showed that Thr7 is more conservative than Thr4 in Firmicutes (Supplementary Fig. 9a, b), so we speculated that the Thr7 may play the key role in CcpS function. It is an interesting phenotype. Of course, we need more additional data to support this view in the future.

We also do some experiments on cooperativity between the phosphorylation of T7 and T4. This data is shown below, but not included in the revised manuscript. Surprisingly, the Thr7 phosphorylation has no significant effect on Thr4 phosphorylation, but non-phosphorylated Thr4 significantly promote phosphorylation on Thr7. Here, we really appreciate the reviewer's good suggestions again, we will further explore these interesting phenomenons in the future.

- lines 162-163: These experiments are not fully convincing. Additional images should be shown. A statistical analysis made with many cells is also necessary. I know that such statistics are difficult with TEM images as it requires many images but maybe the authors should consider using another method (quantification of the fluorescence signal of living cells immuno-detected with an anti-CPS antibody). This approach would also help in demonstrating if cell morphogenesis is affected (based on the analysis of phase contrast images). Related to these points, is the cell shape of the *CpsD-3YF* mutant affected (lines 251-253). This mutation induces cell elongation and delocalization of the CPS production. It is important to conclude that phenotypes caused by *CpsD* and *CppS* are similar (line 252).

Thanks for the good suggestion. We have made statistical analysis with enough cells (Fig. 3h, i) (lines 307-310). In addition, we have compared the cell-associated CPS fluorescence signal and cell morphogenesis. Statistical analysis have made with enough cells. We will include these data in the revised manuscript (Fig. 6g, h). We observed that the CPS production of *ccpS-T4VT7V* mutant strain and *cpsD-3YE* mutant strain were similar and significantly increased compared to WT strain, while *ccpS-T4ET7E* mutant strain showed a similar CPS level with WT strain. Moreover, CPS production of *cpsD-3YF* mutant strain was significantly decreased compared to WT strain and the non-encapsulated Δcps mutant strain no CPS detected (Fig. 6g, h)

(lines 618-623).

In addition, we also explore the effect of CpsD mutant on the cell shape based on DIC, we found that either non-phosphorylated CpsD (*cpsD-3YF*) or its phosphomimetic form (*cpsD-3YE*) strains showed similar cell shape with the WT strain. Interestingly, *cpsD-3YF* mutant strain with aberrant cell elongation and nucleoid defects in *S. pneumoniae*⁶. To the best of our knowledge, the biological characteristics are quite different between *S. pneumoniae* and *S. suis* in many physiological processes. And CpsD phosphorylation acts as a signaling system coordinating CPS synthesis with chromosome segregation was only systematically studied in *S. pneumoniae*^{6,7}. *S. suis* also encodes a typical tyrosine kinase/phosphatase system, CpsD/CpsB⁸. However, their exact regulatory roles in *S. suis* capsule synthesis have not been reported yet. Although we have demonstrated that CpsBCD plays a similar and important role in CPS production in *S. suis* in the manuscript, we can not explain the difference between *S. pneumoniae* and *S. suis* about the above morphogenesis phenomenon, and thus we speculated that this is only the beginning of research in *S. suis*. This data is shown below, but not included in the revised manuscript.

- Line 257: Is CpsD-CppS interaction modulated by CpsD phosphorylation?

Thanks. We also tested the rCpsD-3YE and rCpsD-3YF interactions with rCcpS by SPR, and will include this data in the revised manuscript (Supplementary Fig. 6a-d). Interestingly, we found that rCcpS-T4VT7V also showed higher affinity to rCpsD than rCcpS-T4ET7E. In addition, we found that CpsD phosphorylation did not affect the affinity to CcpS (Supplementary Fig. 6a and c, or b and d) (lines 518-529).

- There are some typos and English errors that should be corrected thorough the manuscript. For instance, line 142, 215, 216, 399-400, ...)

We really appreciate the reviewer's careful reading. We have corrected it.

Minor comments:

- Abstract, lines 18-19: it is already established that CpsB and CpsD are indispensable in Streptococci (and in firmicutes actually). I would change for "we also confirm that...."

Thanks. We have rephrased the complete abstract (lines 11-25).

- Introduction, lines 31-32: the "majority" is actually not fully accurate. I would be better to simply state that there are 2 main systems, the the ABC-transporter dependent system and

wzy system, the latter being the most widespread. Please cite Whitfield et al, 2020, Annu Rev Microbiol.

We thank the referee for pointing this out, and we will revise it and add this reference in the revised manuscript (lines 35-38).

- A series of appropriate references (more recent and/or more related to Streptococci and/or alternative conclusions) are missing: Line 41: please cite Jadeau et al, 2008, Bioinformatics as reference for the BY-kinase family. Line 53: please cite as well Ducret&Grangeasse, 2021, Curr Opin Microbiol. Lines 59-60: Please cite Nagarajan et al, 2022, Trends Microbiol. Line 63: Please cite Fleurie et al, 2014, PLoS Genet. Line 225: Please cite Nourikyan et al, 2015, PLoS Genet.

Thanks. We will add these references as suggested: Jadeau et al, 2008, Bioinformatics as reference for the BY-kinase family (lines 51-53), Nagarajan et al, 2022, Trends Microbiol (lines 65-67), Fleurie et al, 2014, PLoS Genet (lines 74-75), Nourikyan et al, 2015, PLoS Genet (lines 376-379). In addition, we have carefully revised the entire introduction (lines 27-93)

- Supp Fig 2d: Not easy to read. A color code could be helpful.

We are very sorry for the mistake. We have revised it and also other figures in the revised manuscript (Supplementary Fig. 3c).

Reviewer #2 (Remarks to the Author):

The manuscript by Tang and co-workers describes a substantial body of work to understand the regulation of capsule synthesis in *Streptococcus suis*. They report a new link between the Stk1 kinase and capsule production mediated by interaction between the CcpS kinase substrate and the CpsB capsule synthesis regulator. This new connection is of interest and expands the landscape of physiological outputs that are known to be regulated by Stk1 homologs in bacteria.

The most novel and impactful findings reported in this manuscript are the connection between CcpS and CpsB and the resulting influence on CpsBCD phosphorylation and capsule synthesis. Many of the other findings reported here that relate to: (1) Stk1 phosphorylation of CcpS at T4 and T7; (2) impact of mutations at T4 and T7 on CcpS function; (3) the structure of CcpS and the existence of the flexible N-terminal tail; (4) impact of mutations at the dimer interface on CcpS; and (5) alterations of CcpS phosphorylation in response to environmental stimuli (suspension in PBS in particular), all have been reported previously for homologs of Stk1/CcpS in other bacterial species (*Enterococcus* mainly, but also *Listeria* to some extent) and therefore serve mainly to confirm in *S. suis* what has already been observed. Of course it is important in this manuscript to demonstrate similar results experimentally in *S. suis* for these results to be rigorous overall, but the authors never cite the previous work and instead present their findings as though nothing was previously known about this kinase/substrate signaling pair (or the CcpS 'family' of proteins), when that is in fact not the case.

Response: We really appreciate the reviewer's careful reading and good suggestions. The constructive comments and suggestions have helped us improve the quality of our manuscript. As noted by the reviewer, we found that the PASTA kinase Stk1 directly regulates capsule synthesis through the CpsBCD system via a novel regulator CcpS. CcpS is a homologue of ReoM whose function has recently been described in *Listeria*². Interestingly, the mechanism of CcpS function we have proposed here would represent a novel mechanism of regulation not mediated by regulated proteolysis. In addition, the crosstalk between Stk1 and CpsBCD, as well as the novel role for the IreB/ReoM homologue CcpS are unique and important for pathogens.

We are very sorry for the mistake that we did not fully discuss the previous work in our original manuscript. We will further highlight the unique and significance of our study by proper comparison and discussion of our results relative to those of the previous work, and thus make it clearer for readers. We have added related references and further discussions at proper position **in the revised manuscript** (lines 245-270) (lines 790-805).

Line 80-81: The authors refer to a “comparative analysis of phosphoproteins” between wild-type and Stk1-depleted cells that led them to investigate CcpS. However, other than CcpS, no data from this analysis is presented. Was CcpS the only differentially phosphorylated protein identified? If not, it would be of interest to see a list of all the differentially phosphorylated proteins that could be added to the supplemental material.

We apologize for the vagueness of this statement. We identified a potential Stk1 specific substrate hypothetical protein RS00400 (renamed CcpS) by phosphoproteomic analysis in a previous study. We will clarify this point in the revised manuscript (lines 102-104).

CcpS was not the only identified differentially phosphorylated protein in *S. suis*. We detected a phosphorylated version of CcpS-specific peptides, but not the whole sequence of CcpS. We have excluded any additional *in vivo* phosphorylation sites for CcpS by a series of experiments (Fig. 1f, Supplementary Fig. 1f, g). We have observed that Stk1 and CcpS form an interaction complex and the kinase Stk1 specifically phosphorylates CcpS at residues Thr4 and Thr7 (lines 123-142).

Fig 1f (and several other IP figures also): Immunoblotting was performed on IP samples to analyze CcpS phosphorylation, and GroEL immunoblots were included as “loading control”. It is unclear how GroEL could be used as a “loading control” for such samples – was GroEL also IP'd with the anti-CcpS antibody?

We are very sorry for the mistake. In fact, GroEL as loading control for the total CcpS protein level (not shown) from the whole-cell lysates in the same samples, but not for IP. We have corrected it in figure (Fig. 1e, f).

Line 232: There appears to be a typo; it appears the mutations were made in *cpsD*, not *ccpS*. We are very sorry for the mistake. We have revised it (line 388).

Lines 334-335: The authors state that “CcpS phosphorylation level decreased rapidly and maintained a low level over time” in regards to figure 4f, which seems misleading because it appears from the figure that overall phosphorylation level actually remains relatively constant

while the amount of total CcpS protein increases upon PBS challenge. In other words, the existing CcpS may not have necessarily been dephosphorylated after PBS challenge, but rather that new CcpS was synthesized and did not become phosphorylated. Or at the very least, it is not possible to distinguish between these possibilities on the basis of the data presented.

Thanks for the helpful comments. To address this, we also performed Western blot assay to detect total CcpS protein level at the same time. The results showed that the CcpS phosphorylation level of cells decreased rapidly and maintained at a low level over time, while the total CcpS protein level no change (lines 183-187). And we have corrected it in this figure (Supplementary Fig. 2a, b).

Fig 4: The legend title indicates that Stk1/CcpS “improves the stress tolerance of *S. suis*”, but this is not justified because there is no data to demonstrate this.

Thanks for your comments. We have replaced it as “CcpS phosphorylation participates in various stress response” in the revised manuscript (line 229).

Lines 476-477: The authors state that Stk1/CcpS control of capsule synthesis “enhances the survival ability of bacteria against the stress of the host immune system”. This is an overstatement of the data presented that is not justified. While the data do show an effect of Stk1/CcpS on survival in macrophages, whether or not that survival effect is specifically the result of an influence on capsule synthesis has not yet been established experimentally through analysis of appropriate double mutants.

Thanks for your comments. We have rewritten this section in the revised manuscript (lines 808-814).

Line 485: The authors state that “Stk1 is conserved almost in the whole field of bacteria”, which is inaccurate. Homologs of Stk1 – in other words, kinases with extracellular PASTA domains - are almost ubiquitous among Firmicutes and Actinobacteria, but they are rare among Gram-negative bacteria.

Thank you for pointing out this. We have revised it (lines 769-770).

Line 625: It appears there should be a reference here to *S. aureus*, and not *E. faecalis*, based on the data presented.

Thanks. We have corrected it. In addition, construction of plasmids has been summarized in *Supplementary Methods* (lines 616-718).

Experimental conditions for Phos-tag gels (including acrylamide percentage, phos-tag concentration, divalent cation identity and concentration) should be included for all.

Thanks for your comments. We have rewritten this related sentence (lines 963-965).

Lines 795-807 (phagocytosis assay): Have controls been performed to establish that the antibiotic regimen used is sufficient to kill extracellular *S. suis* bacteria? No such controls are described or shown. If this is known from previous work, a citation should be included.

Thanks for your comments. We have rewritten this section accordingly and cited the reference

above (lines1110-1112).

Supplementary figure 1c: Appropriate controls are not included. We cannot rule out the possibility that the band labeled as CcpS-P is actually Stk1.

Thanks. We have replaced the fig by new one (Supplementary figure 1d).

Supplementary figure 1f: I was unable to discern which line corresponded to which sample due to the low resolution (low resolution was an issue for many of the figures). Also, the legend states 3 replicates were performed: are the averages plotted? I do not see any mention of statistical analysis.

We are very sorry for the mistake. We have replaced it with a color code fig (Supplementary figure 1h). In addition, we have checked all figures and ensured high resolution. Statistical analysis has been performed in the revised manuscript.

Supplemental figure 3a: I performed searches at NCBI for several of the gene numbers listed. Although I did not test all genes listed, the results I obtained did not correspond with the annotations given in the table. Please correct.

We apologize for the vagueness of this figure (Supplementary figure 4a). The old version of genome still coexisting with the new one. And the new version of genome GI: 820722437

Reviewer #3 (Remarks to the Author):

Tang et al has illustrated the role of conserved Coordinator of Capsular Polysaccharide Synthesis (CcpS) playing a crucial role in Capsular polysaccharide (CPS) synthesis, which itself it pivotal to virulence. Using biochemical methods, the authors have shown that CcpS is a substrate of serine/threonine kinase Stk1. They have shown CcpS play the role of a regulatory hub in connecting bacterial tyrosine kinase system CpsBCD into the STK.

Response: We really appreciate your constructive and invaluable suggestions. The constructive comments and suggestions have helped us improve the quality of our manuscript. We tried to go over the manuscript carefully and make it clearer for readers, and we have added more experimental data and analysis **in the revised manuscript**.

1. This system is very similar to the IreK-IreB protein regulatory network in gram positive bacteria (reference below). Could the authors highlight any unique differences between them? Hall, C. L., Tschannen, M., Worthey, E. A., & Kristich, C. J. (2013). IreB, a Ser/Thr kinase substrate, influences antimicrobial resistance in *Enterococcus faecalis*. *Antimicrobial agents and chemotherapy*, 57(12), 6179-6186.

Thanks for the helpful comments. We agree with the reviewer's first point. We have demonstrated that the Stk1-CcpS axis is remarkably active in the life course of *S. suis* (Fig. 2) (lines 175-211). In fact, the *S. suis* cells with continuous starvation in PBS showed that the CcpS phosphorylation level of cells decreased rapidly and maintained at a low level over time (Supplementary Fig. 2a, b), which was similar with the previously reported that CcpS homolog

IreB phosphorylation decreased in PBS treatment⁹, and thus suggested that Stk1-CcpS and IreK-IreB system were all important in these bacterial physiology (lines 175-211).

However, we regret for an unclear formulation in the original manuscript. In the manuscript, we found that the PASTA kinase Stk1 directly regulates capsule synthesis through the CpsBCD system via a novel regulator CcpS. CcpS is a homologue of ReoM whose function has recently been described in *Listeria*². Interestingly, the mechanism of CcpS function we have proposed here would represent a novel mechanism of regulation not mediated by regulated proteolysis. In addition, the crosstalk between Stk1 and CpsBCD, as well as the novel role for the IreB/ReoM homologue CcpS are unique and important for pathogens. We apologize for the vagueness of the presentation of the original manuscript again. We have carefully rephrased in the revised manuscript to further highlight the innovation of our work.

2. The above paper also mentions the role of Threonine residues as substrates for STK. Do the authors using molecular modeling or simulation methods show the structural basis for the interaction and catalysis.

Thanks for the good suggestion. The structural basis for the interaction and catalysis is interesting and will provide new insights into the substrates interaction with STK. However, these data will not necessarily germane to the story at hand. Thus, we are very sorry for that these potential interesting further analysis will be not included in the revised manuscript. In fact, we focus on getting more insights into the architecture and mechanism of CcpS modulates the activity of CpsB (lines 481-543) in our work.

In addition, we also do some experiments on cooperativity between the phosphorylation of Thr7 and Thr4. This data is shown below, but not included in the revised manuscript. Surprisingly, the Thr7 phosphorylation has no significant effect on Thr4 phosphorylation, but non-phosphorylated Thr4 significantly promote phosphorylation on Thr7. Here, we really appreciate the reviewer's good suggestions again, we will further explore these interesting phenomenons in the future. And perhaps molecular modeling or simulation methods will provide more insights into the structural basis for the interaction and catalysis.

3. Although the crystal structure is solved for the WT and mutant CcpS, no structural analysis is performed and figure 5F, is not very clear in what authors want to represent.

The justification of structure is not clear nor it gains in new insights into the substrate interaction or the role of mutation. Either a MD simulation or a complex of CcpS with STK1 could shed some more insights.

Thank you for pointing out this point of confusion. We have added the structural analysis at proper position in the revised manuscript (Supplementary figure 1h) (lines 482-495). In addition, we have performed molecular docking for CpsB and CcpS (figure 5c, d). Data

presented here provide a new insight into the molecular mechanism of a protein with functional disorder region controls bacterial physiology. We showed that CcpS and CcpS-T4E7E variant folded structures were highly matched (Fig. 5a). Although electron density map of N-terminus was not clear, the partial folded structure of N-terminus between CcpS and CcpS-T4E7E chain A crystal structure was obviously different (Fig. 5a, chain A, N-ter). Although the exact mechanism by which non-phosphorylated CcpS inhibits CpsB activity is still unknown, analysis of molecular docking and affinity showed that non-phosphorylated CcpS has higher affinity to CpsB much more than CcpS-P (Fig. 5c-f). All the results and analysis have been incorporated in the revised manuscript (lines 496-529) (lines 851-881).

In addition, we thank the reviewer's good suggestions again. Molecular modeling or simulation methods perhaps provide more insights into the structural basis for the interaction and catalysis for Stk1-CcpS complex, as well as the cooperativity between the phosphorylation of Thr7 and Thr4 in the future.

4. Overall, the quality of figures are subpar and it would be great to improve the figures in the further corrections.

Thanks for the suggestive comment. We have checked all the figures and ensured high resolution in the revised manuscript.

References

1. Hall, C.L., Tschannen, M., Worthey, E.A. & Kristich, C.J. IreB, a Ser/Thr kinase substrate, influences antimicrobial resistance in *Enterococcus faecalis*. *Antimicrob Agents Chemother* **57**, 6179-6186 (2013).
2. Wamp, S. et al. PrkA controls peptidoglycan biosynthesis through the essential phosphorylation of ReoM. *Elife* **9** (2020).
3. Du, W. et al. Two active forms of UDP-N-acetylglucosamine enolpyruvyl transferase in gram-positive bacteria. *J Bacteriol* **182**, 4146-4152 (2000).
4. Blake, K.L. et al. The nature of *Staphylococcus aureus* MurA and MurZ and approaches for detection of peptidoglycan biosynthesis inhibitors. *Mol Microbiol* **72**, 335-343 (2009).
5. Kelliher, J.L. et al. PASTA kinase-dependent control of peptidoglycan synthesis via ReoM is required for cell wall stress responses, cytosolic survival, and virulence in *Listeria monocytogenes*. *Plos Pathog* **17** (2021).
6. Nourikyan, J. et al. Autophosphorylation of the Bacterial Tyrosine-Kinase CpsD Connects Capsule Synthesis with the Cell Cycle in *Streptococcus pneumoniae*. *Plos Genet* **11** (2015).
7. Mercy, C. et al. RocS drives chromosome segregation and nucleoid protection in *Streptococcus pneumoniae*. *Nat Microbiol* **4**, 1661-1670 (2019).
8. Smith, H.E. et al. Identification and characterization of the cps locus of *Streptococcus suis* serotype 2: the capsule protects against phagocytosis and is an important virulence factor. *Infect Immun* **67**, 1750-1756 (1999).
9. Labbe, B.D. & Kristich, C.J. Growth- and Stress-Induced PASTA Kinase

Phosphorylation in *Enterococcus faecalis*. *J Bacteriol* **199** (2017).

Reviewer #1 (Remarks to the Author):

The authors have performed significant efforts to address my comments. Even if some differences between their observations and what is described in the literature for other strains remain surprising, I agree that they could reflect specific features of *S. suis* biology as suggested by the authors and that it will be interesting to investigate them in the future. I however believe that some of the data presented in the cover letter should be included in the manuscript, notably the MIC for Δ murZ and Δ murA1 that could be in Sup Fig 3B. Same for Δ ccpS virulence in animal models that could be included in Fig 7. Last, I think that phase contrast images of Δ ccpS presented in the cover letter and showing that deletion of ccpS has no impact on the cell shape, and consequently morphogenesis, should be included in the manuscript (in Figure 3 for instance) and analyzed properly (these images can be analyzed rapidly with MicrobeJ to generate violin plots (for instance)). This is particularly important because I am not super convinced about this claim because the 2 TEM images of Figure 3h suggest aberrant cell shape and aberrant positioning of the division septum (that is not at the cell center) with the presence of mini-cells-like structures.

Reviewer #2 (Remarks to the Author):

In this resubmission, the authors have done a significant amount of work to respond to the comments on the original manuscript. It is the opinion of this reviewer that the authors have addressed the comments in a satisfactory manner. I have only a handful of minor editorial comments listed below to improve the clarity of the resubmitted manuscript.

Lines 16, 83, 209, 787, 805: "remarkably active" or "extremely active". The modifiers "remarkably" and "extremely" are not really justified, because there are no comparisons with any other signaling pathways that would provide context or relative evaluation of activity. Seems sufficient to simply state that the pathway is active, or that its activity changes in response to environmental stimuli etc.

Line 75: "negatively inhibit" seems redundant and actually incorrect. I presume the authors meant "inhibit" or "negatively regulate" or similar.

Line 91: Given that the Stk1-[IreB/ReoM/CcpS] signaling pathway has already been demonstrated in multiple divergent genera of bacteria, I do not think the results in this paper "reveal that the Stk1-CcpS axis constitutes a widespread signaling and regulation mechanism". Perhaps "results are consistent with the idea that ..." or something similar.

Line 207: "oxygen pressures" does not seem correct. "hydrogen peroxide" or similar would be more appropriate.

Line 367: something seems to be missing – cognate what of CpsB? Please clarify.

Reviewer #3 (Remarks to the Author):

The authors have substantially worked on the manuscript to improve it overall and my concerns on the structural analysis and molecular docking is addressed along with other comments by reviewers 1 and 2.

Response to the Reviewers' comments:

Dear Reviewers,

We really appreciate your careful reading and constructive suggestions to improve our manuscript. In this version, we modified the manuscript as recommended. **All main changes were highlighted in red in the revised manuscript.** In addition, the point-to-point responses to the Reviewer's comments are as following:

In summary (main revisions):

1. We have moved images about the MICs for $\Delta murA1$ or $\Delta murZ$ mutant strains (Supplementary Fig. 3h) (lines 202-203), and $\Delta ccpS$ mutant strain virulence in animal models (Supplementary Fig. 9h) (lines 500-501) to our manuscript, as reviewer 1 suggested.
2. We have added statistical analysis (Supplementary Fig. 3i, j) (lines 205-208), as reviewer 1 suggested.
3. We have checked and revised the manuscript according to editorial requests.
4. We addressed all other issues raised by reviewers and made corrections for trivial mistakes. **All main changes were highlighted in red in the revised manuscript.**

Reviewer #1 (Remarks to the Author):

The authors have performed significant efforts to address my comments. Even if some differences between their observations and what is described in the literature for other strains remain surprising, I agree that they could reflect specific features of *S. suis* biology as suggested by the authors and that it will be interesting to investigate them in the future. I however believe that some of the data presented in the cover letter should be included in the manuscript, notably the MIC for $\Delta murZ$ and $\Delta murA1$ that could be in Sup Fig 3B. Same for $\Delta ccpS$ virulence in animal models that could be included in Fig 7. Last, I think that phase contrast images of $\Delta ccpS$ presented in the cover letter and showing that deletion of *ccpS* has no impact on the cell shape, and consequently morphogenesis, should be included in the manuscript (in Figure 3 for instance) and analyzed properly (these images can be analyzed rapidly with MicrobeJ to generate violin plots (for instance)). This is particularly important because I am not super convinced about this claim because the 2 TEM images of Figure 3h suggest aberrant cell shape and aberrant positioning of the division septum (that is not at the cell center) with the presence of mini-cells-like structures.

Response: Thanks for reviewer's support on our manuscript. We have added the above data to our manuscript according to reviewer's suggestion. These include the MICs for $\Delta murA1$ or $\Delta murZ$ mutant strain (Supplementary Fig. 3h) (lines 202-203), $\Delta ccpS$ mutant strain virulence in animal models (Supplementary Fig. 9h) (lines 500-501).

Here, we really appreciate the reviewer's good suggestions for the method. The data mentioned above have been further analyzed, and we found that *ccpS* disruption did not cause an overall change in cell shape, and the overall cell shape of *ccpS-T4ET7E* and *ccpS-T4VT7V* mutant stains were similar with WT strain (Supplementary Fig. 3i, j) (lines 205-208). In addition, we had randomly chosen the field of view in electron microscope, which may

present abnormal morphological observation due to inappropriate viewing angle.
Special thanks to you for your good comments.

Reviewer #2 (Remarks to the Author):

In this resubmission, the authors have done a significant amount of work to respond to the comments on the original manuscript. It is the opinion of this reviewer that the authors have addressed the comments in a satisfactory manner. I have only a handful of minor editorial comments listed below to improve the clarity of the resubmitted manuscript.

Response: We really appreciate your careful reading and constructive suggestions to improve our manuscript. We have revised our manuscript according to reviewer's suggestion.

Lines 16, 83, 209, 787, 805: "remarkably active" or "extremely active". The modifiers "remarkably" and "extremely" are not really justified, because there are no comparisons with any other signaling pathways that would provide context or relative evaluation of activity. Seems sufficient to simply state that the pathway is active, or that its activity changes in response to environmental stimuli etc.

Thanks for your comments. We have corrected these sections in the revised manuscript (lines 79, 174, 538, 557).

Line 75: "negatively inhibit" seems redundant and actually incorrect. I presume the authors meant "inhibit" or "negatively regulate" or similar.

Thank you for pointing out this. We have revised it (line 71).

Line 91: Given that the Stk1-[IreB/ReoM/CcpS] signaling pathway has already been demonstrated in multiple divergent genera of bacteria, I do not think the results in this paper "reveal that the Stk1-CcpS axis constitutes a widespread signaling and regulation mechanism". Perhaps "results are consistent with the idea that ..." or something similar.

Thanks. We have corrected it (line 87).

Line 207: "oxygen pressures" does not seem correct. "hydrogen peroxide" or similar would be more appropriate.

Thanks. We have replaced it (line 172).

Line 367: something seems to be missing – cognate what of CpsB? Please clarify.

**We are very sorry for the mistake. We have corrected it in the revised manuscript (line 278).
Once again, thank you very much for your comments and suggestions.**

Reviewer #3 (Remarks to the Author):

The authors have substantially worked on the manuscript to improve it overall and my concerns on the structural analysis and molecular docking is addressed along with other comments by reviewers 1 and 2.

Response: Thanks for reviewer's great support on our manuscript.